# The Arabidopsis phosphatase PP2C12 negatively regulates LRX-RALF-FER-mediated cell wall integrity sensing

Xiaoyu Hou [1], Kyle W Bender [1], Álvaro D Fernández-Fernández [1], Gabor Kadler [1,4], Shibu Gupta [1,5], Mona Häfliger [1], Amandine Guérin [1], Anouck Diet [2], Stefan Roffler[1], Daniela Campanini [1], Thomas Wicker [1], Cyril Zipfel [1,3] & Christoph Ringli [1✉]

## Abstract

**Plants have evolved an elaborate cell wall integrity (CWI) sensing system to monitor and modify cell wall formation. LRR-extensins (LRXs) are cell wall-anchored proteins that bind RAPID ALKALI-NIZATION FACTOR (RALF) peptide hormones and induce compaction of cell wall structures. LRXs also form a signaling platform with RALFs and the transmembrane receptor kinase FERONIA (FER) to maintain cell wall integrity. *LRX1* of *Arabidopsis thaliana* is predominantly expressed in root hairs, and *lrx1* mutants develop defective root hairs. Here, we identify a regulator of LRX1-RALF-FER signaling as a suppressor of the *lrx1* root hair phenotype. The *repressor of lrx1_23* (*rol23*) gene encodes PP2C12, a clade H phosphatase that interacts with FER and dephosphorylates Thr696 in the FER activation loop in vitro. Expression of FER phospho-mimetic and phospho-null mutants in an *lrx1 fer-4* background demonstrates that phosphorylation of FER at Thr696 is essential for suppression of *lrx1* phenotypes by *rol23*. The LRX1-related function of PP2Cs appears clade H-specific. Collectively, our data suggest that LRX1 acts upstream of the RALF-FER signaling module and that PP2C12 inhibits FER via activation-loop dephosphorylation.**

**Keywords** LRX1; PP2C12; FERONIA; Arabidopsis; Root Hairs
**Subject Category** Plant Biology

## Introduction

The plant cell is surrounded by the cell wall (CW) that protects the cell from biotic and abiotic stresses and, importantly, provides tensile strength to counteract the intracellular turgor pressure (Ali et al, 2023). The controlled expansion of the CW is a prerequisite for cell growth to take place. This process requires CW remodeling and wall material synthesis in a controlled manner (Höfte and Voxeur, 2017). To achieve that, plants have evolved an elaborate sensing system that involves CW proteins and cell membrane-associated receptor kinases (RKs) to monitor the dynamics of CW structures, and to orchestrate the intracellular activities to maintain cell wall integrity (CWI) (Cheung and Wu, 2011; Wolf et al, 2012; Wolf and Höfte, 2014; Hamann, 2015).

In *Arabidopsis thaliana* (hereafter, Arabidopsis), the RK FERONIA (FER) has been identified as a major regulator of a plethora of cellular processes to coordinate temporal and spatial cell expansion and CW enlargement (Escobar-Restrepo et al, 2007; Duan et al, 2010; Wolf and Höfte, 2014; Ji et al, 2020). *FER* belongs to the *Catharanthus roseus* Receptor-like kinase 1-like (CrRLK1L) gene family that has 17 members in Arabidopsis and is conserved across the plant kingdom. Several other CrRLK1L proteins, like CURVY1, HERCULES 1/2 (HERK1/2), ANXUR1/2 (ANX1/2), BUPS1/2, THESEUS1, and ERULUS also contribute to CWI sensing (Hematy and Hofte, 2008; Lindner et al, 2012; Boisson-Dernier et al, 2013; Haruta et al, 2014; Ge et al, 2017; Schoenaers et al, 2018). This family of proteins is characterized by a cytosolic protein kinase domain, a single-pass transmembrane domain and an extracellular malectin-like domain (Franck et al, 2018). FER is a receptor for plant RAPID ALKALINIZATION FACTOR peptide hormones (RALFs) that were identified as developmental and stress regulatory factors with the ability to induce apoplastic alkaliniza-tion and cytosolic $Ca^{2+}$ fluxes, and to alter protein-protein interaction dynamics of RLKs (Pearce et al, 2001; Haruta et al, 2014; Stegmann et al, 2017). A recent study has shown that RALFs form condensates with de-esterified pectic fragments to bind FER, and as such to activate FER-mediated signaling (Liu et al, 2024; Rößling et al, 2024).

Leucine-rich repeat extensins (LRXs) are CW proteins that are crucial for the structure and mechanical properties of CWs (Baumberger et al, 2003; Fabrice et al, 2018; Moussu et al, 2023; Schoenaers et al, 2024). They contain an N-terminal LRR domain

[1]Department of Plant and Microbial Biology, University of Zurich and Zurich-Basel Plant Science Center, Zurich, Switzerland. [2]Institute of Plant Sciences Paris-Saclay (IPS2), CNRS, INRAe, Univ d'Evry, Université Paris-Cité; Université Paris-Saclay, Orsay, France. [3]The Sainsbury Laboratory, University of East Anglia, Norwich Research Park, Norwich, United Kingdom. [4]Present address: University Hospital Zurich, Department of Intensive Care, Zurich, Switzerland. [5]Present address: Securecell AG, Urdorf, Switzerland.
✉E-mail: chringli@botinst.uzh.ch

that forms dimers and binds RALF peptides with high affinity (Mecchia et al, 2017; Herger et al, 2019; Moussu et al, 2020). Recent studies showed that LRX-RALF complexes interact with de-esterified pectins in a charge-dependent manner to condense CW structures (Moussu et al, 2023; Schoenaers et al, 2024). The C-terminal extensin domain with typical features of hydroxyproline-rich glycoproteins anchors the LRX proteins to the CW matrix (Rubinstein et al, 1995; Baumberger et al, 2001; Ringli, 2010). The Arabidopsis genome encodes 11 LRXs; LRX7-11 are expressed in the reproductive tissue while LRX1-6 are expressed in the vegetative tissue, among which LRX1 and LRX2 are predominantly expressed in root hairs (Baumberger et al, 2001; Baumberger et al, 2003; Baumberger et al, 2003a). In vegetative tissue, higher-order mutants of LRXs are phenotypically similar to the fer-4 knock-out mutant, and accumulating evidence suggests that LRXs, RALFs, and FER function in a signaling pathway necessary for salt stress tolerance, the coordination of intra- and extracellular processes necessary for the growth process, and immune-responses (Zhao et al, 2018; Dünser et al, 2019; Herger et al, 2020; Gronnier et al, 2022).

While diverse functions are attributed to the LRX/RALF/FER activities, the regulation of FER is not yet well understood. Type 2 C phosphatases (PP2Cs) are the largest phosphatase family with 80 members in Arabidopsis that are classified in 13 clades (Xue et al, 2008; Fuchs et al, 2013) and are involved in the regulation of different signaling pathways. In Arabidopsis, clade A PP2Cs and PP2C12 of clade H are involved in modulating abscisic acid (ABA)-mediated signal transduction (Allen et al, 1999; Park et al, 2009; Santiago et al, 2009; Weiner et al, 2010; Bai et al, 2020), whereas clade D members act in regulating auxin-induced cell expansion and salt tolerance (Spartz et al, 2014; Fu et al, 2023). The clade C member AP2C1 targets mitogen-activated protein kinases (MAPKs) to regulate immune and hormonal outputs (Schweighofer et al, 2007). POLTERGEIST-LIKE 4 (PLL4), PLL5, PP2C38 and PP2C15 negatively regulate pathogen-associated molecular pattern (PAMP)-triggered immunity (PTI). PLL4 and PLL5 were recently identified as negative regulators shared by developmental and immune receptor signaling (Couto et al, 2016; DeFalco et al, 2022; Diao et al, 2024). Moreover, the clade A PP2C, ABI2, is engaged in the ABA-RALF signaling crosstalk by attenuating the phosphorylation status of FER (Chen et al, 2016).

To gain insights into the regulatory network of the LRX/RALF/FER signaling module, a screen was performed on the lrx1 mutant background to identify mutants suppressing the lrx1 root hair formation defect (Baumberger et al, 2001). One such suppressor mutant, repressor of lrx1_23 (rol23), is an allele of PP2C12. PP2C12 not only affects root hair development but, together with the other members of PP2Cs of the same clade, PP2C15 and PP2CH3, modulates RALF1 sensitivity of seedlings and generally root growth in a FER-dependent manner. Additionally, PP2C12 interacts with and dephosphorylates FER, suggesting that it might attenuate FER protein kinase activity. Together, the data suggest that activating FER suppresses the lrx1 mutant phenotype, an effect that is induced by removing the FER activity-inhibiting PP2C12. This gives further support to the scenario of an LRX/RALF/FER signaling module that influences CW formation and cell growth.

# Results

## Mutations in ROL23 suppress the lrx1 root hair defect

The lrx1 mutant is characterized by frequently short, collapsed, and branched root hairs compared to the long, unbranched root hairs of wild-type Col (Fig. 1A,C,D) (Baumberger et al, 2001). An ethyl methanesulfonate mutagenesis screen was performed on the root hair defective mutant lrx1. In the M2 generation, repressor of lrx1 (rol) mutants that restore the root hair formation in lrx1 were selected, leading to the identification of rol23 that suppresses lrx1 and induces development of wild type-like root hairs both in terms of developmental defects as well as elongation (Fig. 1A,C,D).

A backcross of lrx1 rol23 with lrx1 resulted in F1 seedlings developing an lrx1 root hair phenotype (Fig. 1A), indicating that rol23 is a recessive mutation. The rol23 mutant was analyzed by whole-genome sequencing (WGS, details see Methods). The sequencing data of lrx1 rol23 was compared to the existing WGS data of lrx1 (Schaufelberger et al, 2019) and SNPs were identified in several linked genes on chromosome 1 (Appendix Fig. S1A). The mutation causing the rol23 phenotype was identified in a segregation analysis using molecular markers for the different SNPs on seedlings of the F2 population described above, combined with crossing T-DNA insertion lines of the genes containing SNPs into the lrx1 background. The T-DNA insertion allele of At1g47380, a gene encoding the PP2C-type phosphatase PP2C12 (Xue et al, 2008), caused suppression of lrx1 (Fig. 1A). The T-DNA allele causes interruption of PP2C12 at the codon for Glu174 while rol23 is a missense mutation leading to a Gly66Arg substitution (Fig. 1B). An additional allele containing the insertion of one base pair in the codon Ile31, thus inducing a shift in the ORF, was produced by CRISPR/Cas9-mediated gene editing and also caused suppression of lrx1 (Fig. 1A–D; Appendix Fig. S1C). These independent alleles of PP2C12 all suppressing lrx1 demonstrated that rol23 is an allele of PP2C12. PP2C12 belongs to the clade H of PP2Cs (Appendix Fig. S1B), together with PP2C15 and a third PP2C that has an inconsistent numbering in the literature [(Xue et al, 2008) versus (Diao et al, 2024)] and is thus named PP2C of clade H_3 (PP2CH3) (Table 1). Based on single-cell RNA sequencing data (Ryu et al, 2019), all three PP2CH family members are expressed in root hairs, with PP2CH3 showing a lower expression than PP2C12 and PP2C15 (Appendix Fig. S1D). The T-DNA allele described above is referred to as pp2c12-2 [pp2c12-1 is a T-DNA insertion mutant described in (Bai et al, 2020)], the CRISPR/Cas9-induced mutation as pp2c12-3 (Fig. 1B; Appendix Fig. S1C). rol23 also induces root hair formation in the almost root hair-less lrx1 lrx2 double mutant (Baumberger et al, 2003; Herger et al, 2020) (Appendix Fig. S1E).

A complementation assay was performed by expressing GFP-tagged PP2C12 under its native promoter (PP2C12::PP2C12-GFP) in the lrx1 rol23 double mutant. Seedlings of independent transgenic lines display an lrx1-like root hair defect, further confirming that the mutation in PP2C12 causes the suppression of lrx1. It is noteworthy that the T2_3 line with the strongest accumulation of PP2C12-GFP (Fig. 2B,D) shows an over-complementation of the lrx1 rol23 phenotype with strongly reduced root hair formation resembling an enhanced lrx1 root hair defect comparable to fer-4 (Fig. 3A).

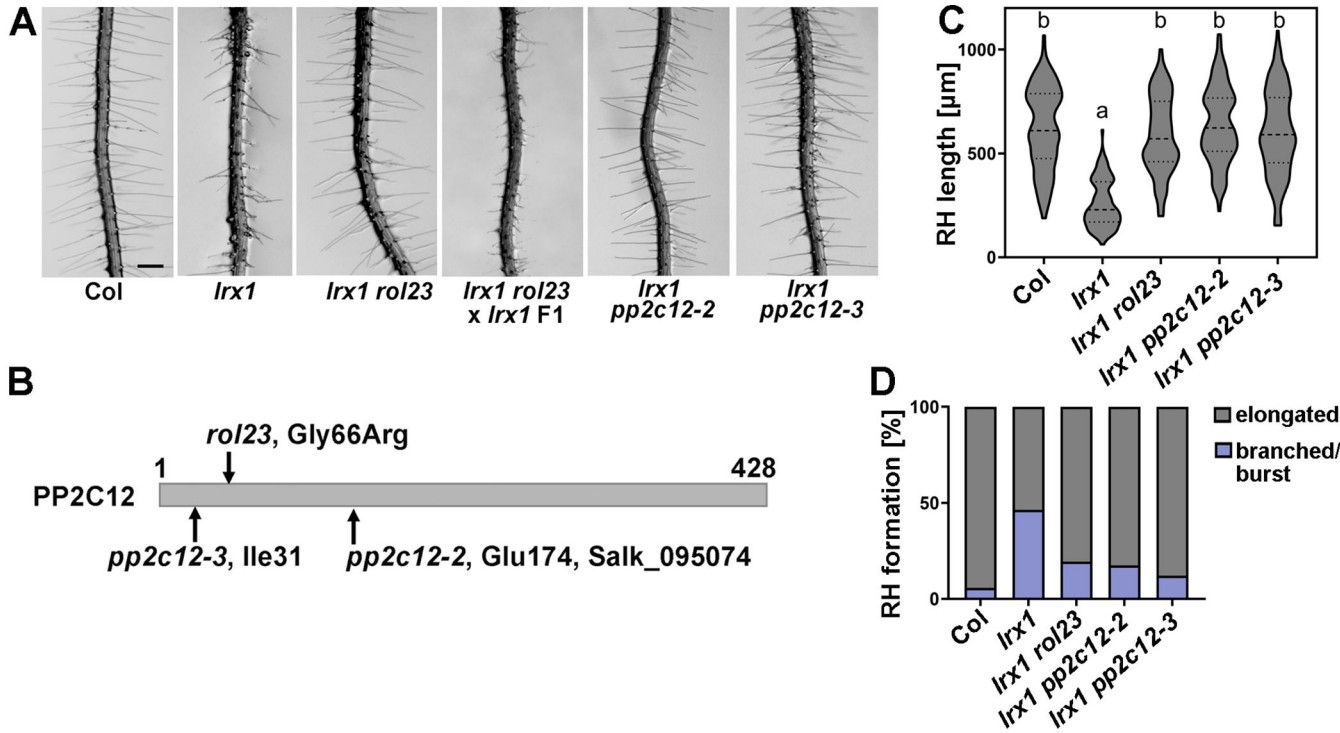

**Figure 1. Mutations in *PP2C12* suppress *lrx1*.**

(A) Representative images of 5-day-old vertically grown Arabidopsis seedlings. The *lrx1* mutant develops defective root hairs compared to the wild-type (Col), which is suppressed by *rol23* and the *pp2c12-2* and *pp2c12-3* alleles. Scale bar = 500 μm. (B) Schematic diagram of the 428 AA-long PP2C12 protein showing the positions corresponding to the EMS allele *rol23*, T-DNA insertion line *pp2c12-2* (SALK_095074C) and the CRISPR/Cas9-induced allele *pp2c12-3*. (C) Quantification of root hair length (violin plot, the central dash line represents the median, and the dotted lines represent 25% and 75% percent). Genotypes with like letter designations are not statistically different ($n > 100$ per genotype, one-way ANOVA with Tukey's unequal N-HSD post hoc test, $p < 0.01$). All $p$ values can be found in Appendix Table S3. (D) Stacked bar plot showing classification of root hairs with different shapes in the different lines. Source data are available online for this figure.

**Table 1. List of PP2Cs used in this study.**

| Name | Gene identifier | Clade |
|---|---|---|
| PP2C12 | At1g47380 | H |
| PP2C15 | At1g68410 | H |
| PP2CH3 | At1g09160 | H |
| PP2C35 | At3g06270 | L |
| PP2C38 | At3g12620 | D |
| PP2C52 | At4g03415 | E |
| EGR1/PP2C34 | At3g05640 | E |

Clade classification following (Fuchs et al, 2013).

PP2C12-GFP accumulates not only in root hairs but in many cell types in the root where it is found in the cytoplasm, at the plasma membrane (Xu et al, 2025), as well as in the nucleus (Appendix Fig. S1F). The relevance of nuclear localization for the LRX1-related function of PP2C12 was tested with PP2C12 constructs containing a nuclear-exclusion signal (NES) and a mutant version of NES (nes) (Shen et al, 2007). This was done in the context of the *lrx1 lrx2 rol23* mutant. Several independent transgenic lines for both constructs revealed nuclear exclusion of PP2C12-GFP-NES but not PP2C12-GFP-nes. Both variants, however, equally complemented the *lrx1 lrx2 rol23* phenotype

(Appendix Fig. S1F), indicating that nuclear localization of PP2C12 is not essential for its activity in the context of LRX1-mediated root hair development.

## Functional specificity within the clade H PP2Cs

Functional specificity of PP2Cs was tested by complementation of *lrx1 rol23* with *PP2C12::PP2Cs-GFP* using different *PP2Cs* from the clade H and other clades (Table 1). Lines with comparable GFP fluorescence were analyzed (Fig. 2A,C,D; Appendix Fig. S2). While *PP2C15* and, to a slightly reduced extent, *PP2CH3*, restored the *lrx1* root hair phenotype, i.e., complemented *rol23* (Fig. 2), *PP2Cs* of other clades did not have the same effect. This suggests a distinct function and activity in the LRX1-related process of clade H PP2Cs compared to other PP2Cs.

## *rol23* suppresses the *fer-5* root hair defect

As LRXs are hypothesized to function coordinately with FER (Zhao et al, 2018; Dünser et al, 2019; Herger et al, 2020; Gronnier et al, 2022), we tested the effect of *rol23* on the *fer* knock-out and knock-down mutant alleles *fer-4* and *fer-5* (Duan et al, 2010), respectively. While *fer-4* is essentially root hair-less, *fer-5* has a T-DNA insertion downstream of the protein kinase coding sequence and exhibits an intermediate root hair defect (Fig. 3A,B). The root hair defect in

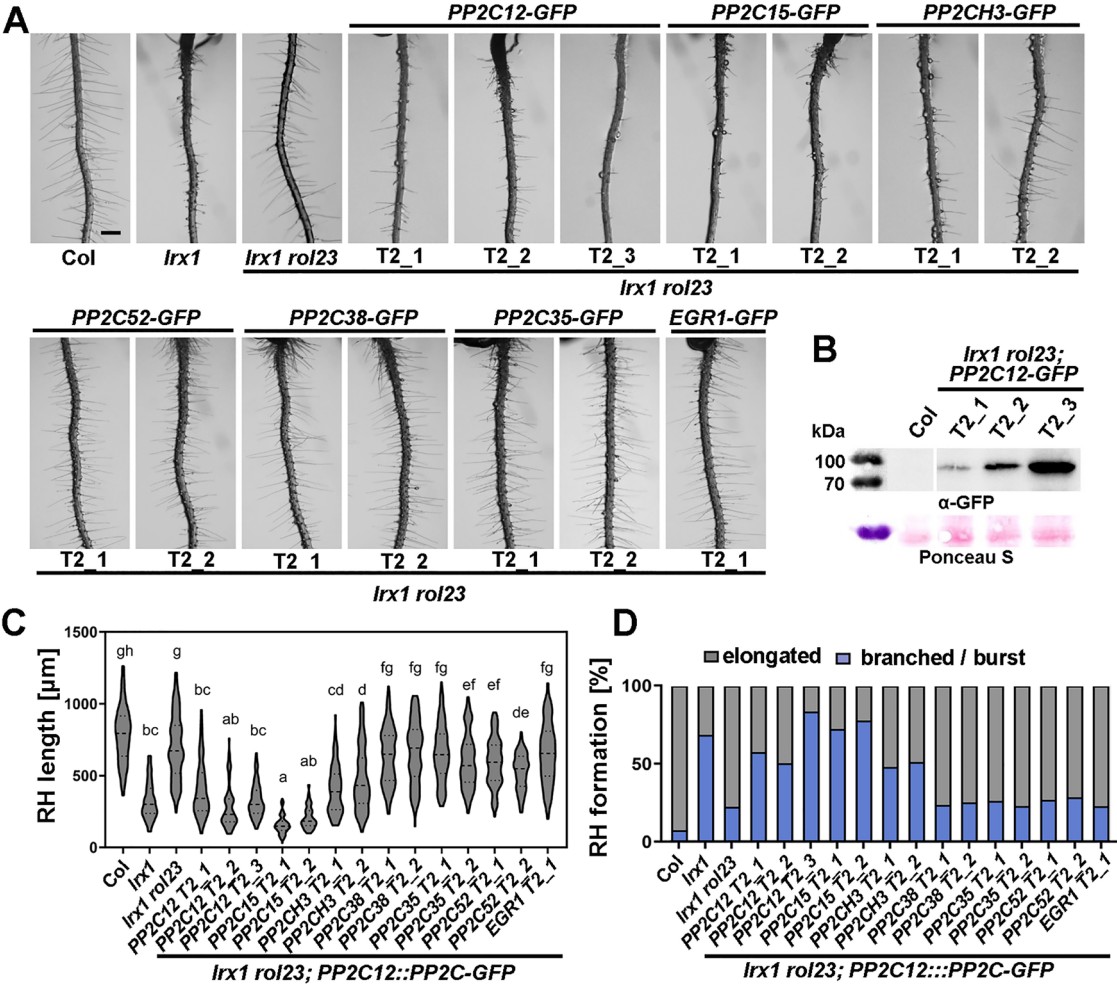

**Figure 2. Clade H *PP2Cs* can complement the *rol23* phenotype.**

(A) Representative images of 5-day-old vertically grown Arabidopsis seedlings. Complementation of *lrx1 rol23* with *PP2C12::PP2C-GFP* constructs in independent transgenic lines reveal complementation of *rol23* by *PP2C12*, *PP2C15*, and *PP2CH3*, but not by *PP2Cs* of other clades. Scale bar = 500 μm. (B) Immunoblot with anti (α)-GFP antibody on total extracts of seedlings of *lrx1 rol23; PP2C12-GFP* T2 lines, with T2_3 showing the highest protein accumulation and over-complementation of the *lrx1 rol23* phenotype as shown in (A). Membrane stained with Ponceau S shows comparable loading. (C) Statistical analyses of root hairs shown in (A), with the violin plot showing the elongated root hair length (the central dash line represents the median, and the dotted lines represent 25% and 75% percentiles). Genotypes with like letter designations are not statistically different (*n* > 100 per genotype, one-way ANOVA with Tukey's unequal N-HSD post hoc test, *p* < 0.01). All *p* values can be found in Appendix Table S3. (D) Stacked bar plot showing classification of root hair phenotypes in the different lines (*n* > 80). Source data are available online for this figure.

*fer-5* is significantly suppressed by the *rol23* mutation (Fig. 3A,B). By contrast, *rol23* does not suppress the root hair phenotype of the knock-out mutant *fer-4* (Fig. 3A). Additional mutations in PP2C15 and PP2CH3 induced by CRISPR/Cas9 (Appendix Fig. S1C) did not alter root hair development in *fer-4* (Appendix Fig. S3A). This finding suggests that PP2C12 influences FER function in root hair development.

PP2C15 was shown to modify BAK1 activity, and our data show that PP2C12 and PP2C15 have overlapping activities. Hence, we wanted to rule out the possibility that BAK1 is important for the *lrx1* and *rol23* phenotypes. To this end, the knock-out *bak1-4* and the *bak1-5* alleles (Heese et al, 2007; Schwessinger et al, 2011) were crossed with *lrx1 rol23*. The selected *lrx1 bak1-4* and *lrx1 bak1-5* double mutants show an *lrx1*-like phenotype, and the *lrx1 rol23 bak1-4* and *lrx1 rol23 bak1-5* triple mutants show an *lrx1 rol23* root hair development (Fig. 3A,C), ruling out a role of BAK1 in this process.

## Clade H PP2Cs negatively regulate RALF1 sensitivity via FER

Mutants lacking functional *FER* and *LRX* genes are impaired in the sensitivity towards RALF peptides (Haruta et al, 2014; Mecchia et al, 2017; Stegmann et al, 2017; Zhao et al, 2018; Dünser et al, 2019; Abarca et al, 2021) (Fig. 3D). We assessed root growth inhibition and found that RALF1 sensitivity in *rol23*, *pp2c12-2*, and in *rol23 pp2c15 pp2ch3* remains unchanged compared to Col. The *fer-4* and *fer-5* mutants showed no and intermediate RALF1 sensitivity, respectively. While the RALF1 sensitivity of *fer-5* was not altered in *fer-5 rol23*, it was increased in the *fer-5 rol23 pp2c15 pp2ch3* line (Fig. 3D,E). By contrast, neither *rol23* nor *rol23 pp2c15 pp2ch3* alter RALF1 insensitivity of *fer-4* (Fig. 3D,E). Interestingly, *PP2C12-GFP* overexpression lines showed significantly reduced sensitivity to RALF1 (Fig. 3D; Appendix

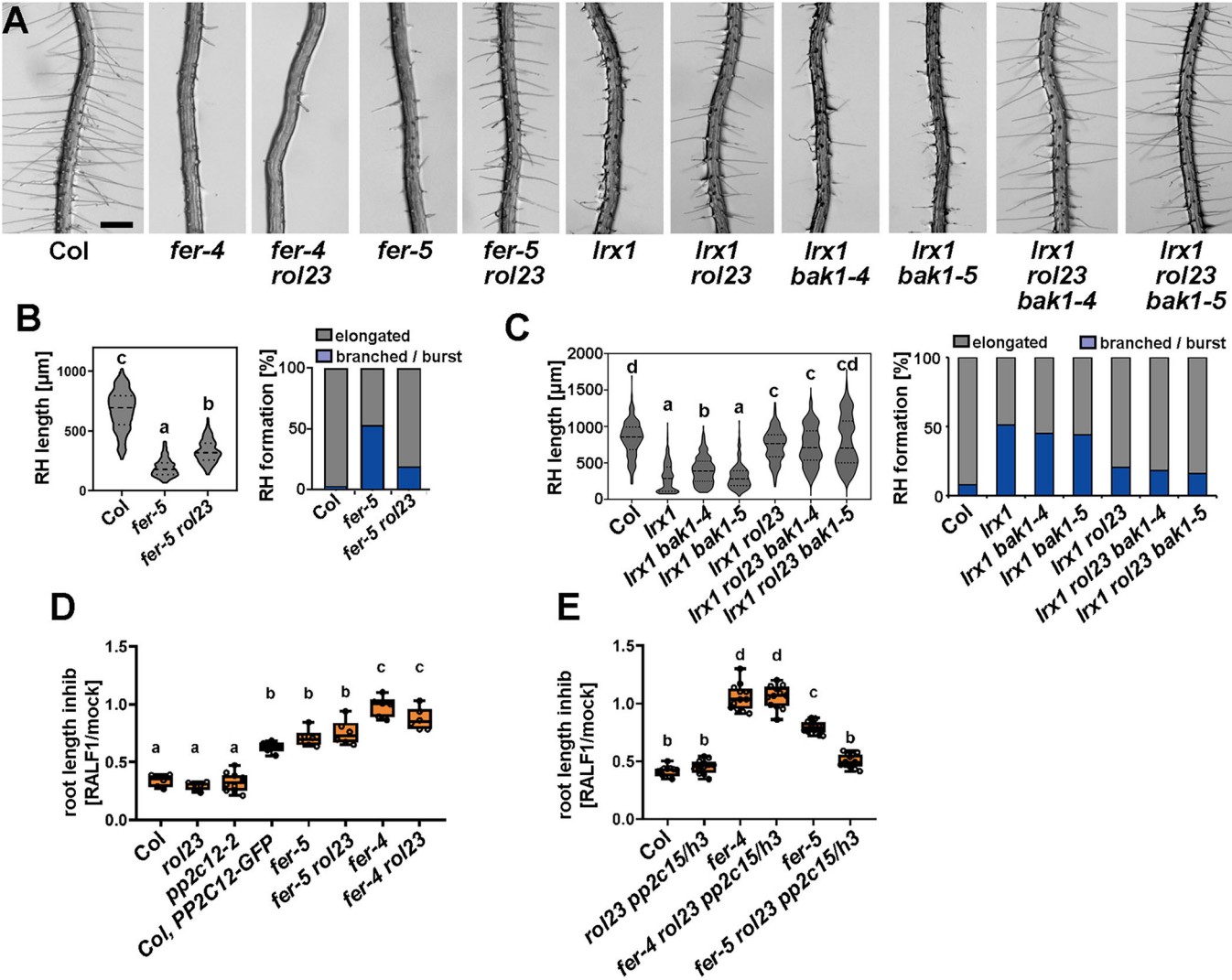

**Figure 3. PP2C12 influences FER-related processes.**

(A) Roots of 5-day-old Arabidopsis seedlings grown in a vertical orientation. The knock-down allele *fer-5* is suppressed by *rol23*, but the root hair-less *fer-4* knock-out allele is not affected. *bak1* mutations have no influence on *lrx1* or the suppression of *lrx1* by *rol23*. Scale bar = 500 μm. (B, C) Quantification of root hairs shown in (A). Violin plot shows the elongated root hair length (the central dash line represents the median, and the dotted lines represent 25% and 75% percentiles) and the stacked bar plot shows classification of root hairs with different shapes of different lines (*n* > 100 per genotype). Genotypes with like letter designations are not statistically different (*n* > 100 per genotype, one-way ANOVA with Tukey's unequal N-HSD post hoc test, *p* < 0.01). All *p* values can be found in Appendix Table S3. (D, E) Quantification of primary root length of 7-day-old seedlings grown in the absence (mock) or presence of 2 μM RALF1 peptide. The primary root length is shown as relative to mock data for each genotype. Genotypes with like letter designations are not statistically different (one-way ANOVA with Tukey's unequal N-HSD post hoc test, *p* < 0.01). All *p* values can be found in Appendix Table S3. The boxes represent 25% and 75% percentiles, the central line the median, and the whiskers the minimum and maximum values, respectively. Similar results with at least two independent experiments were obtained. Source data are available online for this figure.

Fig. S3B–D), which was not observed when testing *e.g.* a PP2C35-GFP line that showed comparable protein accumulation (Appendix Fig. S3C,D). The *PP2C12-GFP* overexpression lines used to demonstrate reduced RALF1 sensitivity also developed a root hair phenotype with frequently enlarged root hair bases but intact root hair elongation (Appendix Fig. S3E–G), indicating that excessive PP2C12 modifies root hair development in otherwise wild type plants. Together, this data suggests that the PP2Cs of clade H negatively influence RALF1 sensitivity, and this effect is dependent on functional FER.

## The PP2C12 phosphatase activity is required for its function in LRX1-mediated signaling pathway

To assess if the catalytic activity of PP2C12 is required to establish the root hair defect in *lrx1*, a PP2C12 variant was produced where two Asp residues essential for phosphatase activity and conserved among PP2Cs (Fig. 4A) (Jackson et al, 2003; Pan et al, 2015; DeFalco et al, 2022) are mutated to Asn. The transgenic *lrx1 rol23* plants expressing catalytically inactive *PP2C12-GFP*, referred to as *PP2C12*<sup>dead</sup>-*GFP*, under the *PP2C12* promoter failed to complement

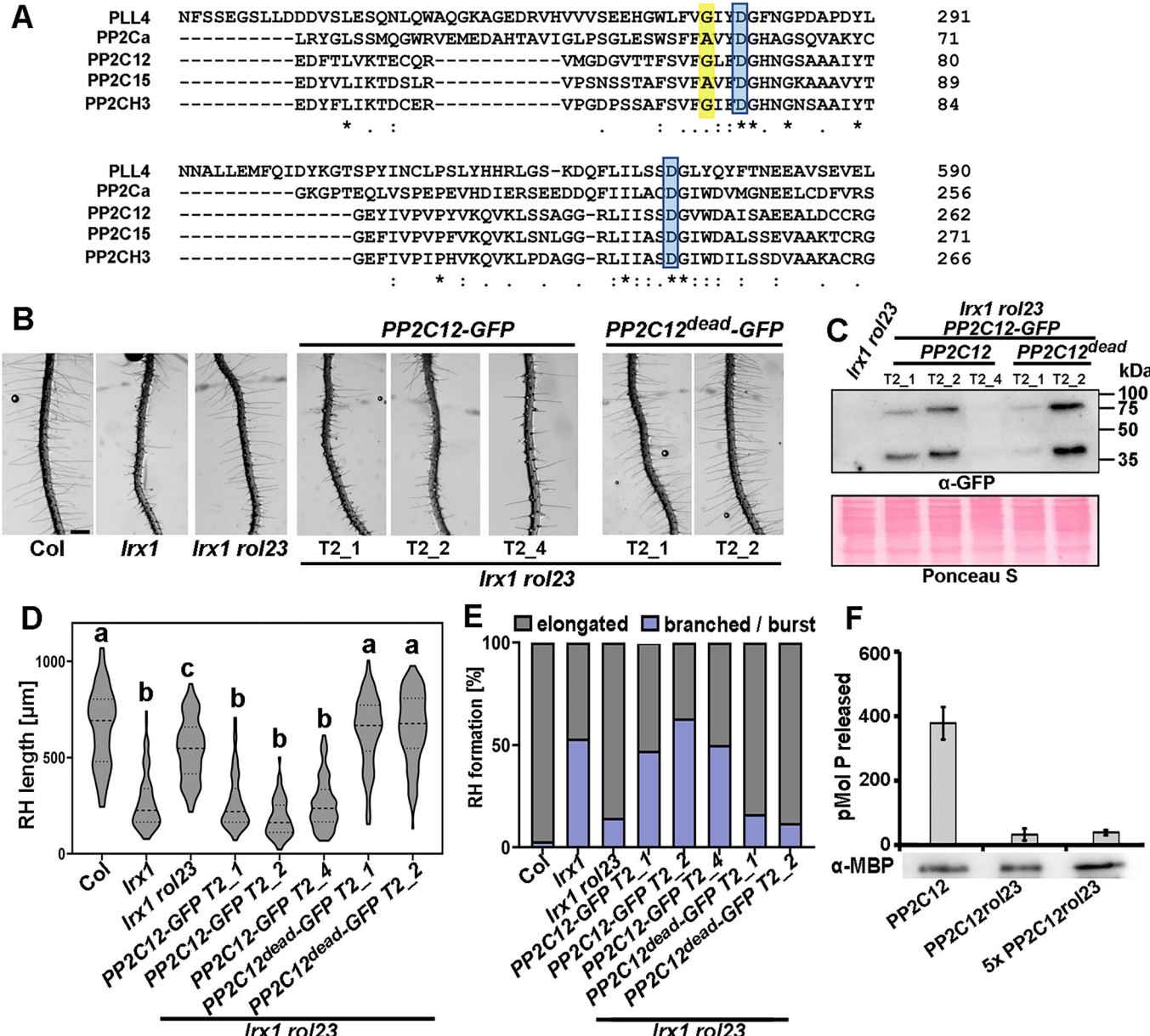

Figure 4. Phosphatase activity of PP2C12 is essential for its function in LRX1-mediated root hair formation.

(A) Protein sequence alignment of PP2C12, PP2C15, and PP2CH3 with the Arabidopsis PLL4 and the human PP2Cα shows the conservation of the Asp residues (boxed in blue) that are essential for phosphatase activity. The position of the Gly66Arg substitution on PP2C12 is highlighted in yellow. Sequence alignment was done with Clustal Omega https://www.ebi.ac.uk/jdispatcher/msa/clustalo. Identical, conserved, and similar positions in the alignment are indicated by asterisks, colons, or single dots, respectively (B) Representative images of 6-day-old Arabidopsis seedlings grown in a vertical orientation. The PP2C12dead variant with the two Asp substituted by Asn are not able to complement the lrx1 rol23 phenotype. Several independent T2 generations are shown. Scale bar = 500 μm. (C) Immunoblot probed with anti(α)-GFP antibody on total extracts of seedling roots shown in (B). Blot stained with Ponceau S shows comparable loading. (D) Quantification of the root hair phenotype shown in (B). In the violin plot, the median is represented by the central dash line, and the 25% and 75% percentiles are represented by the dotted lines (n > 100 per genotype). Genotypes with like letter designations are not statistically different (one-way ANOVA with Tukey's unequal N-HSD post hoc test, $p < 0.01$). All $p$ values can be found in Appendix Table S3. (E) The stacked bar plot represents the classification of different root hair types in each genotype. Different letters indicate significant differences. (F) Phosphatase activity of MBP-PP2C12 and MBP-PP2C12rol23 proteins. $n \geq 3$ per activity value, bars indicate SD. An anti-MBP immunoblot on the protein extracts used for the phosphatase assay confirms comparable amounts of recombinant protein at the expected size of around 90 kDa. Source data are available online for this figure.

rol23 despite protein accumulation comparable to transgenic lines containing PP2C12-GFP (Fig. 4B–E). This demonstrates that the catalytic activity of PP2C12 is essential for its function in LRX1-mediated root hair development. Based on these findings, we tested

whether the PP2C12 with Gly66Arg substitution encoded by rol23 (referred to as PP2C12rol23) is affected in its catalytic activity. To this end, both PP2C12 and PP2C12rol23 were expressed as fusion proteins with a maltose-binding protein, purified from E. coli, and

catalytic activity was tested (details see in Methods). Using comparable amounts of recombinant proteins, the catalytic activity of PP2C12$^{rol23}$ was strongly reduced compared to PP2C12. This remained low, even if a five-fold excess of PP2C12$^{rol23}$ was used (Fig. 4F). The Gly66Arg substitution induced by the *rol23* mutation is close to the first of the two Asp (D) residues important for Mg$^{2+}$/Mn$^{2+}$ metal binding and thus the catalytic activity (Pan et al, 2015) (Fig. 4A). Gly66 is a residue not entirely conserved among phosphatases, with the human PP2Cα and PP2C15 having an Ala at this position. This suggests that a non-polar nature (Gly or Ala) is important at this position and a positively charged Arg residue encoded in the *rol23* mutant is not tolerated.

## PP2C12 interacts with and dephosphorylates FER

Our genetic analysis suggests that PP2C12 is a negative regulator of LRX-RALF-FER mediated signaling. To further understand the physiological connection between PP2C12 and the LRX-RALF-FER signaling module, we tested the physical association between FER and PP2C12 in planta. In stable transgenic Arabidopsis, we observed co-immunoprecipitation of FER with PP2C12-GFP (Fig. 5A). The interaction of phosphatases with their substrates is often transient, explaining the rather weak detection of FER. Therefore, we used PP2C12$^{dead}$-GFP for a co-immunoprecipitation assay with FER-FLAG expressed in *N. benthamiana*. This revealed an association between the two proteins, while FER-FLAG did not bind Lti6b-GFP (Fig. 5A,B) or GFP (Appendix Fig. S4A) used as negative controls. Furthermore, a bimolecular fluorescence complementation (BiFC) assay in *N. benthamiana* showed that PP2C12 interacts with FER but not with Lti6b (Fig. 5C). The same BiFC experimental setup performed with the plasma membrane marker protein REM1.3-mRFP confirmed FER-PP2C12 interaction at the plasma membrane (Appendix Fig. S4B). The discrepancy between the clearly visible BiFC-assay signal and the rather weak Co-IP experiment in Arabidopsis might be explained by the dissociation of PP2C12 from FER during the long incubation times in the immunoprecipitation procedure while the BiFC-induced fluorescence is directly detected. To confirm localization of PP2C12, a virtual section of PP2C12-GFP fluorescence with the plasma-membrane marker FM4-64 was performed (Appendix Fig. S4C), suggesting that PP2C12-GFP is also at the plasma membrane.

Since the phosphatase activity of PP2C12 is required for its function in root hair development (Fig. 4), the dephosphorylation of FER by PP2C12 was tested. To this end, an in vitro phosphatase assay with recombinant proteins expressed in bacteria was performed. The cytoplasmic domain of FER (FERCD)-His, which is auto-phosphorylated during expression in *E. coli*, was incubated with active MBP-PP2C12 and inactive MBP-PP2C12$^{dead}$. Phosphorylation was investigated by immunoblotting using an anti-pThr antibody. The FERCD phosphorylation level was modestly reduced after 5 min and 90 min of incubation with PP2C12 but not PP2C12$^{dead}$ (Fig. 6A), suggesting that PP2C12 dephosphorylates FERCD. In a next step, samples of FERCD incubated with PP2C12 or PP2C12$^{dead}$ were subjected to phosphoproteomic analysis. Quantification of all detectable FERCD phosphopeptides (Appendix Table S1) indicated decreased abundance of a triply phosphorylated peptide carrying phosphorylation at Thr692, Ser695, and Thr696 (Fig. 6B; Appendix Fig. S5A). Consistently, we observed an increase in abundance of a doubly phosphorylated peptide carrying phosphorylation at Thr692 and Ser695 (Fig. 6B;

Appendix Fig. S5B), suggesting that PP2C12 dephosphorylates Thr696 in the FER activation loop (Nolen et al, 2004; Kong et al, 2023). Altered abundance of these two peptides was not observed in samples treated with phosphatase-dead PP2C12$^{dead}$.

## FER Thr696 affects FER activity in planta

To investigate the significance of FER$^{T696}$ phosphorylation on FER activity in planta, *FER::FER-GFP* constructs with either wild-type *FER* (*FER$^{T696}$*), phospho-dead *FER$^{T696A}$*, or phospho-mimetic *FER$^{T696D}$* and *FER$^{T696E}$* were produced and transformed into the *lrx1 fer-4* mutant background. If Thr696 was the only substrate of PP2C12 on FER, a suppression of the *lrx1* root hair defect would be expected, in addition to the complementation of the *fer-4* mutant phenotype. Several independent transgenic lines accumulating different levels of recombinant FER-GFP were identified. For all the constructs, complementation of the *fer-4* mutant phenotype was observed (Fig. 7A), indicating that the substitutions on the Thr696 residue do not have a detrimental effect on FER in vivo. Quantification of root hair formation and elongation revealed that the phospho-dead *FER$^{T696A}$* was less effective than the wild-type *FER$^{T696}$* at complementing the *lrx1 fer-4* phenotype, even if *FER$^{T696A}$*-GPF was much more abundant (Fig. 7C,D). The FER$^{T696D}$ and FER$^{T696E}$ variants show root hair phenotypes comparable to the wild-type FER$^{T696}$, suggesting that the phospho-mimetics do not substitute for phosphorylation at Thr696. Alternatively, Thr696 is not the only substrate of PP2C12 and pT696 is not sufficient to confer suppression of *lrx1*. To test whether pT696 is necessary for suppressing *lrx1*, *PP2C12* was mutagenized by CRISPR/Cas9 in one transgenic line expressing wild-type *FER$^{T696}$*-GFP and two independent lines expressing the phospho-dead *FER$^{T696A}$*-GFP. After introducing the *pp2c12$^{crispr}$* mutations, the line expressing wild-type *FER$^{T696}$*-GFP showed strongly enhanced root hair formation, i.e., suppression of the *lrx1* root hair phenotype. In both *FER$^{T696A}$*-GFP expressing lines, the introduced *pp2c12* mutation showed only slightly improved root hair growth compared to the same T2 lines with wild-type *PP2C12*, despite clearly higher levels of *FER$^{T696A}$* -GFP than FER$^{T696}$-GFP (Fig. 7B,E,F). This suggests that FER pT696 is important for suppression of *lrx1* and Thr696 might not be the only substrate of PP2C12 on FER.

## Clade H members have an impact on FER endocytosis

Ligand-induced endocytosis of cell-surface receptors plays a crucial role in maintaining the duration, amplitude, and specificity of signal transduction activity (Claus et al, 2018). Upon RALF perception, FER undergoes clathrin-mediated endocytosis (Zhao et al, 2018; Yu et al, 2020; Liu et al, 2024). Due to the involvement of clade H PP2Cs in the LRX-RALF-FER signaling pathway, we speculated that they might affect the endocytic dynamics of FER. Therefore, we examined FER-GFP endocytosis in the *fer-4 FER-GFP rol23 pp2c15 pp2ch3* mutant background and observed that FER-GFP was constitutively endocytosed (Fig. 8A,B). This supports the conclusion that clade H PP2C members influence FER-related cellular events. The endocytic tracer FM4-64 did not show different intracellular staining between the two lines (Appendix Fig. S6A), suggesting that the increased endocytosis is rather selective for FER-GFP. Since the methylation status of pectin has a major influence on RALF- and LRX-related processes (Moussu et al, 2023; Rößling et al, 2023; Liu et al, 2024; Schoenaers et al, 2024), we investigated if the enhanced endocytosis of FER in the *fer-4 FER-GFP rol23 pp2c15 pp2ch3* line is affected by the pectin status. The

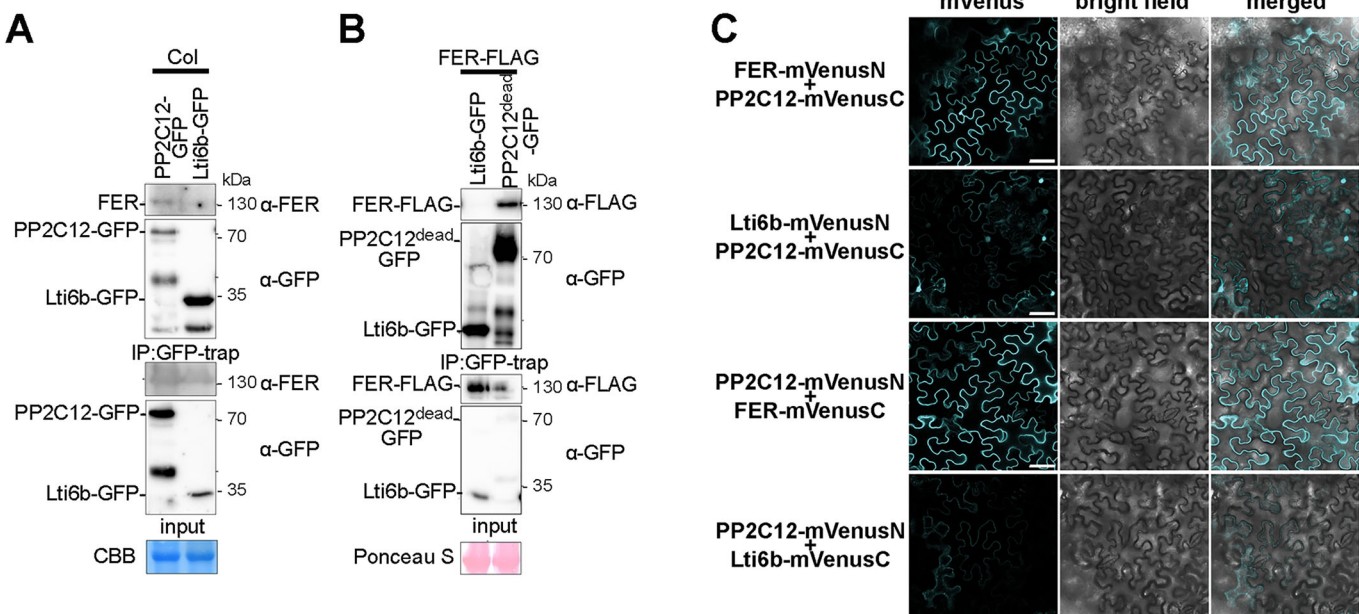

**Figure 5. Interaction of PP2C12 with FER.**

(A) FER interacts with PP2C12 in Arabidopsis. Fourteen-day-old seedlings expressing PP2C12-GFP or Lti6b-GFP were used for immunoprecipitation with anti-GFP trap, followed by detection of FER by an anti(α)-FER antibody, which was only detectable in the line expressing PP2C12-GFP. Blot stained with Coomassie brilliant blue (CBB) shows comparable loading. (B) FER interacts with PP2C12dead in *N. benthamiana*. *FER-FLAG* was transiently co-expressed in *N. benthamiana* leaves with either *PP2C12dead-GFP* or *Lti6b-GFP*. Immunoprecipitation of PP2C12dead-GFP and the PM-marker Lti6b-GFP was done with -GFP trap. FER-FLAG was detected with an anti-FLAG antibody when co-expressed with PP2C12dead-GFP but not Lti6b-GFP. Blot stained with Ponceau S shows comparable loading. (C) BiFC assay of interaction between FER and PP2C12 in *N. benthamiana*. Confocal images show that transient expression of PP2C12 and FER with VenusN or VenusC in both combinations resulted in fluorescent signals at the periphery, whereas little to no signal was observed when replacing FER by the plasma membrane marker Lti6b. The first column shows the fluorescent signal, middle column the bright field image and the right column the merging of the first two columns. Scale bar = 50 μm. Source data are available online for this figure.

amount of de-methylated pectin is reduced in the seedlings by the application of the PME (pectin methylesterase)-inhibitor epigallocatechin gallate (EGCG) (Lewis et al, 2008). *fer-4 FER-GFP rol23 pp2c15 pp2ch3* seedlings treated with 50 μM EGCG showed notably reduced FER-GFP endocytosis (Fig. 8A,B), suggesting that the effect of clade H PP2C members on FER endocytosis is dependent on the methylation status of pectin.

In a next step, the effect of RALF1 on endocytosis of FER-GFP was tested in the *fer-4 FER-GFP* and *fer-4 FER-GFP rol23 pp2c15 pp2ch3* lines (Appendix Fig. S6A). RALF1 induced endocytosis in *fer-4 FER-GFP* as previously shown (Rößling et al, 2023). The *fer-4 FER-GFP rol23 pp2c15 pp2ch3* line shows an increased steady-state level of FER-GFP endocytosis that is not further increased by RALF1 treatment. Finally, we investigated the effect of RALF1 on PP2C12-GFP localization. RALF1 induces membrane invaginations as previously described (Rößling et al, 2023). GFP fluorescence was observed around these invaginations, but other clear differences in PP2C12-GFP distribution were not observed (Appendix Fig. S6B).

## Clade H PP2Cs redundantly regulate root skewing in a FER-dependent manner

While growing the *rol23 pp2c15 pp2ch3* mutant vertically on a hard agar surface, we noticed that the roots skew to the right (Fig. 9A,B). A more detailed analysis revealed ascending primary root skewing in *rol23 pp2c15* < *rol23 pp2ch3* < *rol23 pp2c15 pp2ch3*, while the *rol23,*

*pp2c15*, *pp2ch3*, or *pp2c15 pp2ch3* mutants were indistinguishable from the wild type (Appendix Fig. S7A,B), suggesting that the clade H PP2Cs redundantly influence root skewing, among which PP2C12 and PP2CH3 are the major players. The root skewing phenotype resembles the previously reported right-handed spiral mutants (Furutani et al, 2000) and the *fer-4* mutant has been reported to influence this phenotype (Li et al, 2020). Conceivably, the root skewing phenotype in *rol23 pp2c15 pp2ch3* might rely on FER function. The root skewing of the *fer-5* mutant is not significantly affected, and the *fer-5 rol23 pp2c15 pp2ch3* exhibits skewing comparable to *rol23 pp2c15 pp2ch3* (Fig. 9A,B). *fer-4 rol23 pp2c15 pp2ch3* seedlings show root growth randomly deviating from the gravitropic vector comparable to *fer-4* (Dong et al, 2019). Complementation with a *FER::FER-GFP* construct (Fig. 9A,B) results in root skewing typical for *rol23 pp2c15 pp2ch3*, indicating that the clade H PP2Cs regulate root skewing in a FER-dependent manner. The *fer-4*-induced deviation from the wild-type growth pattern was also observed in our experiments, and this effect is reproduced in *PP2C12-GFP* overexpression lines (Appendix Fig. S7C). Thus, comparable to the RALF1 sensitivity assay, overexpression of *PP2C12-GFP* induces a *fer-4*-like effect.

## Discussion

There is increasingly detailed insight into the process of cell growth and expansion, and a number of RKs have been identified to play a

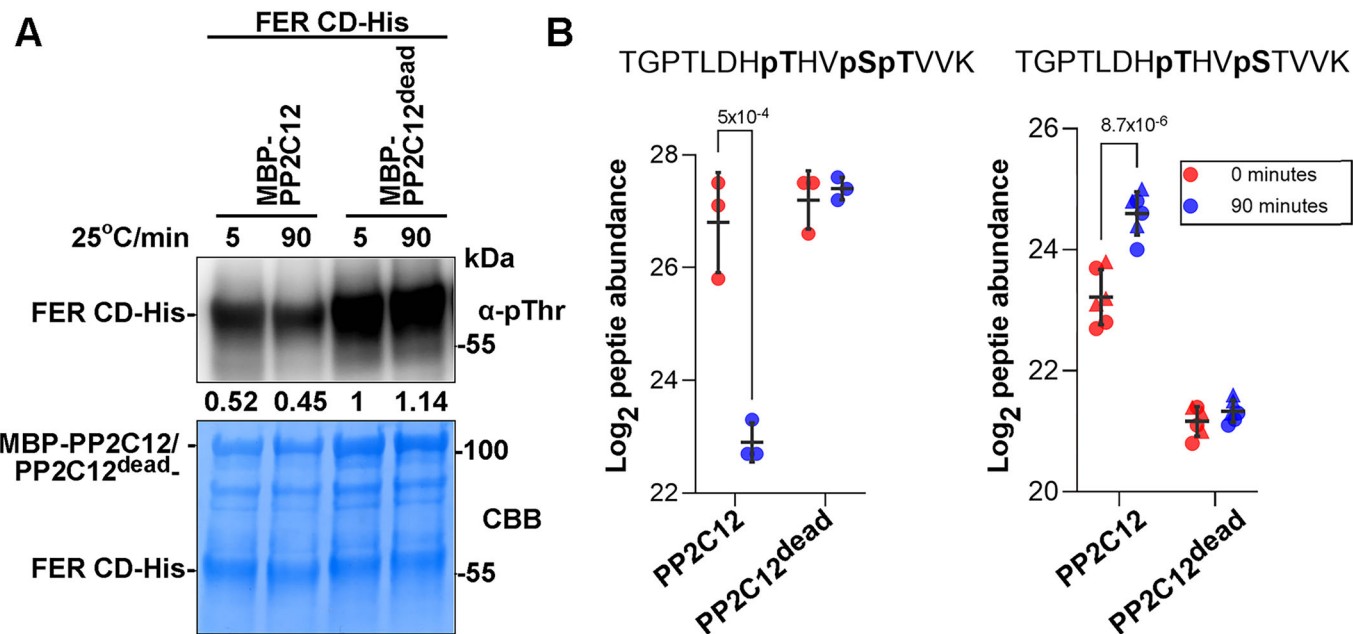

**Figure 6. PP2C12 dephosphorylates FER in vitro.**

(A) MBP-tagged PP2C12, PP2C12$^{dead}$ and His-tagged FERCD (cytoplasmic domain of FER) were expressed in *E. coli*. Purified proteins were incubated as indicated and immunoblotting with an anti($\alpha$)-pThr antibody revealed reduction of FER phosphorylation with PP2C12 but not the phosphatase-inactive PP2C12$^{dead}$. Lower panel shows Coomassie Blue (CBB) staining of the same blot shown in the top panel. (B) Phosphoproteomic analysis with the same sample sets as in (A) revealed a reduction in phosphorylated Thr696 (pT, left panel) and a concomitant increase in non-phosphorylated Thr696 (right panel) upon treatment with PP2C12 but not PP2C12$^{dead}$ after 90 min. Circles and triangles represent doubly and triply charged precursor ions, respectively. *p*-values for statistical differences are shown (Two-way ANOVA, Bonferroni adjustment for multiple comparisons, *n* = 3 or 6). Error bars represent SD. Source data are available online for this figure.

role in CWI sensing, which is essential for controlled cell growth (Kohorn, 2016; Voxeur and Hofte, 2016; Van der Does et al, 2017; Feng et al, 2018). LRX proteins from different tissues have been shown to play relevant roles in CW formation, including the recent findings of their involvement in CW compaction (Baumberger et al, 2003; Draeger et al, 2015; Fabrice et al, 2018; Moussu et al, 2023; Schoenaers et al, 2024). At the same time, LRXs, in conjunction with FER, were identified as playing a role in CWI sensing, salt stress, immune responses, and as high affinity binding sites for RALF peptides (Mecchia et al, 2017; Zhao et al, 2018; Dünser et al, 2019; Herger et al, 2020; Gronnier et al, 2022). While the details of these different processes involving LRX proteins are not fully understood, the work presented here supports the view that LRX1 is involved in a signaling process with RALFs and FER, since modifying the intracellular signaling component PP2C12 can suppress the *lrx1* mutant root hair phenotype.

## Clade H PP2C proteins negatively regulate FER

The biochemical, phosphoproteomic and genetic analyses suggest that the ability of *rol23/pp2c12* mutants to promote root hair growth in *lrx1* is dependent on functional FER. While clade H *pp2c* mutants suppress the intermediate root hair, root skewing, and RALF1 sensitivity phenotypes of the hypoactive *fer-5*, they have no observable impact on the *fer-4* knock-out allele (Figs. 3 and 9), indicating a requirement for FER in the suppression of root hair defects. PP2C12 binds FER and catalyses dephosphorylation at Thr696 in the activation loop, suggesting that the suppression of

root hair defects in *pp2c12* mutants derives from the release of FER (or FER-5) inhibition. Given that phosphorylation of Thr696 was shown to increase FER kinase activity (Kong et al, 2023), our data suggest that PP2C12 might function to directly inhibit FER activation through activation loop dephosphorylation. The finding that the phospho-dead version *FER^T696A* prevents the *pp2c12* mutation to suppress *lrx1* (Fig. 7) further supports this model. Suppression of *lrx1* is induced by increased levels of pT696 in *pp2c12/rol23* mutants, which is prevented in *FER^T696A*. In line with this, overexpression of PP2C12 has an inhibitory effect on FER (Figs. 2 and 3; Appendix Fig. S7). While the overexpression of PP2C12-GFP in the *lrx1 rol23* mutant causes an over-compensation resulting in a root hair-less phenotype (Fig. 2), it only causes a mild root hair formation defect in the wild-type background (Appendix Fig. S3E–G). Thus, the balance between LRX1 function and FER activity is important for regular root hair formation. The characterization of the FER^T696 variants suggests that alteration of Thr696 phosphorylation is an entry point for modifying FER activity, but is not essential for FER, since all phospho-dead and phospho-mimetic variants show complementation of the *fer-4* induced root hair defect and, thus, FER activity.

## (De-)phosphorylation events are common modulators of protein activities

(De-)phosphorylation exerted on and mediated by phosphatases and kinases, respectively, is a major control mechanism to prevent unintended signaling activation and fine-tuning of the signaling

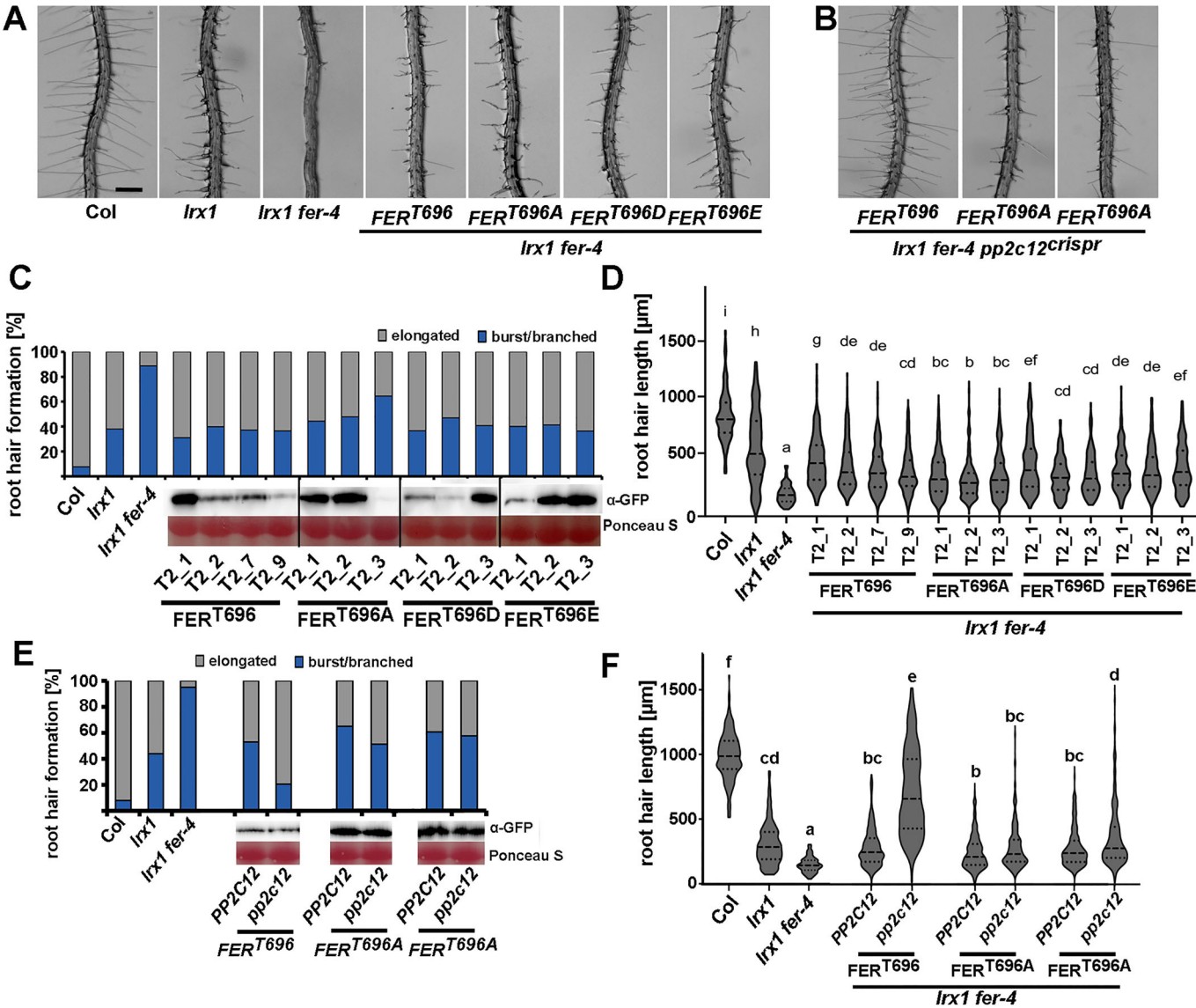

**Figure 7. pT696 of FER is necessary for suppression of _lrx1_.**

_FER_ variants were transformed as _FER::FER-GFP_ constructs into _lrx1 fer-4_ mutants and independent T2 lines were analyzed. (**A**) Root hair phenotypes of T2 seedlings expressing _FER::FER-GFP_ variants as indicated. (**B**) CRISPR/Cas9 mutagenesis of _FER^{T696}-GFP_ and _FER^{T696A}-GFP_ transgenic lines results in suppression of _lrx1_ with FER^{T696} but not _FER^{T696A}_. (**C**) Quantification of root hair formation in the different transgenic lines. Protein abundance was determined by immunoblotting using an α-GFP antibody, equal loading of the gels was confirmed by Ponceau S staining. (**D**) Quantification of root hair length in the same T2 lines. In the violin plot, the median is represented by the central dash line, and the 25% and 75% percentiles are represented by the dotted lines. Genotypes with like letter designations are not statistically different (one-way ANOVA with Tukey's unequal N-HSD post hoc test, _p_ < 0.01, _n_ > 250). All _p_ values can be found in Appendix Table S3. (**E**) Quantification of root hair formation in the transgenic lines and after mutating _PP2C12_ (labeled _PP2C12_ and _pp2c12_, respectively). Immunoblotting using an α-GFP antibody confirmed comparable abundance of FER-GFP in _PP2C12_ and _pp2c12_ backgrounds. Equal loading of the gels was confirmed by Ponceau S staining. (**F**) Quantification of root hair length in the same T2 lines shown in (**E**). In the violin plot, the median is represented by the central dash line, and the 25% and 75% percentiles are represented by the dotted lines. Genotypes with like letter designations are not statistically different (one-way ANOVA with Tukey's unequal N-HSD post hoc test, _p_ < 0.01, _n_ > 250). All _p_ values can be found in Appendix Table S3. Scale bar (**A, B**) = 500 μm.

outputs (Bender and Zipfel, 2023). Several studies have highlighted the important roles of phosphatases, especially type 2C phosphatases, in the negative regulation of RLKs (Li et al, 1999; Shah et al, 2002; Mora-García et al, 2004; Hua et al, 2012; Segonzac et al, 2014; Macho et al, 2015; Couto et al, 2016; DeFalco et al, 2022; Diao et al, 2024). The clade A PP2C phosphatase ABI2 was identified to moderate FER-mediated ABA signaling by dephosphorylating FER

(Chen et al, 2016). Several phospho-sites on the kinase domain and C-term (CT) tail of FER have been identified and shown to be required for the activation of its downstream signaling events (Nolen et al, 2004; Kong et al, 2023). In the current work, we have identified a PP2C12 target residue that is subject to FER autophosphorylation. Yet, the RALF1-induced FER phosphorylation in the CT tail at Ser871 and Ser874 (Haruta et al, 2014) was not observed

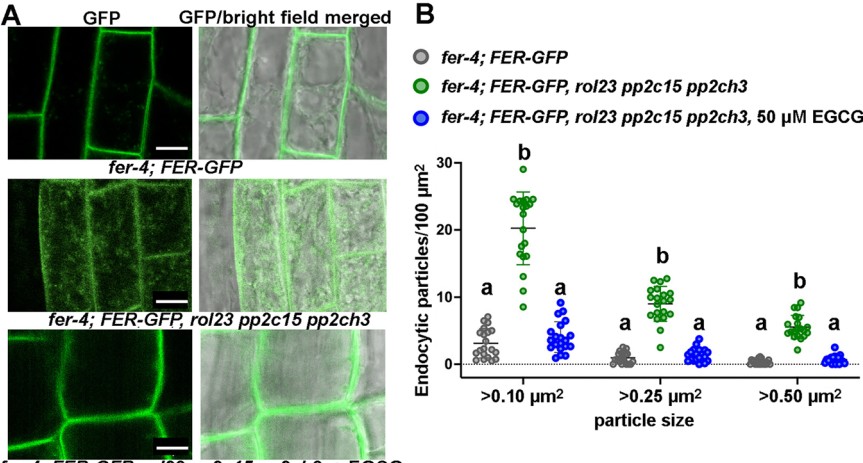

**Figure 8. Clade H PP2Cs influence FER endocytosis.**

(A) Six-day-old Arabidopsis seedlings expressing FER-GFP were used for confocal analysis of endocytosis. FER-GFP internalization is augmented in the *fer-4 FER-GFP rol23 pp2c15 pp2ch3* triple mutant compared to *fer-4 FER-GFP* as shown for the root cells in the transition zone. This effect is reduced by incubating the seedling with the pectin methylesterase inhibitor EGCG for 3 days. Scale bar = 10 µm. (B) Quantification of the cytoplasmic particle sizes in the images as shown in (A) confirm increased endocytosis in the clade H *pp2c* mutant background. Genotypes with like letter designations are not statistically different (*n* = 20, one-way ANOVA with Tukey's unequal N-HSD post hoc test, *p* < 0.01, error bars represent SD). All *p* values can be found in Appendix Table S3. Particles of different sizes in the cells from the root transition zones were quantified. For each treatment/transgenic line, four cells from five seedlings were included. Source data are available online for this figure.

in our assay. Thus, it is possible that residues phosphorylated by other kinases are subject to dephosphorylation by PP2C12.

PP2C12 and PP2C15 have been demonstrated to exert an inhibitory function on the transcription factor ABA-INSENSITIVE 4 (ABI4) and the RK BRI1-ASSOCIATED KINASE 1 (BAK1), respectively (Bai et al, 2020; Diao et al, 2024), revealing that the substrate proteins of these PP2Cs are not restricted to RLKs, but can be different types of proteins influencing apparently different processes. Very recently, PP2CH3 and PP2C12 were identified as a modifier of the Al-binding receptor kinase PSKR1/ALR1 resulting in modulation of Al-induced signaling (Xu et al, 2025). These targets are in different subcellular compartments, explaining the localization of PP2C12 in the cytoplasm, at the plasma membrane, as well as the nucleus (Appendix Figs. S1, S4, S6). Hence, clade H PP2Cs target different proteins and thus potentially provide a molecular link required by the plant to coordinate different processes. It remains to be determined whether additional target proteins of PP2C12 are involved the *rol23/pp2c12*-induced suppression of *lrx1* and the root growth phenotypes caused by the clade H *pp2c* triple mutant. It is possible that LRX1 combines other regulators in addition to FER for the fine-tuning of root hair development, such as additional CrRLK1Ls involved in root hair development like ANX1/2 and ERULUS (Boisson-Dernier et al, 2015; Schoenaers et al, 2018). This hypothesis is in line with the previous finding that the receptor-like cytoplasmic kinase *MARIS*, whose mutation suppresses *anx1 anx2*, also suppresses the *lrx1 lrx2* double mutant phenotype (Boisson-Dernier et al, 2015; Herger et al, 2020).

Phase-separated RALF-pectin condensates trigger the clustering of RALF-cognate and non-cognate cell-surface receptors to activate FER-mediated signaling, along with various other signaling activities mediated by the clustered receptors. As a result, the cell initiates the endocytosis process to moderate the active status of signaling (Yu et al, 2020; Liu et al, 2024). This process is dependent on the pectin status, as RALF-triggered endocytosis of FER is

inhibited by treatment with the low-molecularweight PME inhibitor EGCG and in transgenic lines overexpressing a PME Inhibitor protein (Rößling et al, 2023; Liu et al, 2024). Our results suggest that the clade H PP2Cs contribute to the modulation of FER endocytosis in an EGCG-dependent manner (Fig. 7), but do not influence endocytosis in general, based on FM4-64 staining (Appendix Fig. S6) further supporting that this clade of proteins is important for the regulation of FER activity downstream of RALF/pectin perception by FER. Given that PP2C12 dephosphorylates FER, we speculate that the endocytosis of FER is dependent, at least in part, on its phosphorylation status. On the other hand, RALF1 treatment does not cause an observable re-localization of PP2C12-GFP (Appendix Fig S6). Hence, PP2C12 is a fine-tuning regulator of FER activity that influences the sensitivity of FER to RALF1, explaining the observation that PP2C12 overexpressors are less sensitive to RALF1 (Fig. 3; Appendix Fig. S3).

In conclusion, we identified a negative regulator of LRX1/RALF/FER-mediated signaling during root hair development and root growth, further strengthening that LRX1 is functionally connected with RALF-FER in a signaling process. PP2C12 negatively regulates the LRX1-RALF-FER network, very likely through direct dephosphorylation of FER at the plasma membrane where signals are perceived, subsequently leading to the downregulation of FER activity. Recent work of different groups has revealed functions of LRXs in regulating CW structures (Moussu et al, 2023; Schoenaers et al, 2024) and in participating in signaling processes that relate information on CW status and stresses to the cytoplasm (Zhao et al, 2018; Dünser et al, 2019). Our work suggests that in the *lrx1* mutant, FER activity is reduced, but increased again by the *rol23/pp2c12* mutations, resulting in suppression of *lrx1*. LRXs are strongly interacting with the CW, yet are also part of a connection to the plasma membrane, suggesting they are involved in the dynamic movements of the protoplast relative to the CW during

cell growth and development. Thus, it will be important to investigate whether the signaling and mechanical functions are separate activities of LRX1 or rather are mechanistically linked.

# Methods

### Reagents and tools table

| Reagent/Resource | Reference or Source | Identifier or Catalog Number |
|---|---|---|
| **Experimental models** | | |
| *Arabidopsis thaliana* | Columbia, TAIR | |
| *Arabidopsis thaliana* | various mutants | |
| *Nicotiana benthamiana* | | |
| *E. coli*, BL21 (DE3) | | |
| **Recombinant DNA** | | |
| pMDC111_FER::FER-GFP | Escobar Restrepo et al | |
| pGPTV-Kan | Becker et al, 1992 | |
| pART27 | Gleave, 1992 | |
| pDEST | Invitrogen | |
| pDONR | Invitrogen | |
| pVENUS_N/C | Gehl et al, 2009 | |
| pET28a(+) | EMD Biosciences | |
| pMALc4e | Walker et al, 2010 | |
| pKI1.1 | Tsutsui and Higashiyama, 2017 | |
| pAGM55261 | Grützner et al, 2021 | |
| **Antibodies** | | |
| Anti-GFP | Biolegend | 902601 |
| Anti-mouse HRP | Sigma-Aldrich | A4416 |
| Anti-FLAG HRP | Sigma-Aldrich | A8592 |
| Anti-GFP trap | Chromotek | GTA-20 |
| Anti-FER | Xiao et al, 2019 | |
| Anti-MBP | NE Biolabs | E8032S |
| **Oligonucleotides and other sequence-based reagents** | | |
| PCR primers | This study | Appendix Table S2 |
| **Chemicals, Enzymes and other reagents** | | |
| Restriction enzymes | NEB | |
| LR recombinase | Invitrogen | |
| RALF1 peptide, synthetic | PhtdPeptides | |
| Phosphatase assay kit | Promega | V2460 |
| **Software** | | |
| ImageJ | | |
| **Other** | | |
| PVDF membranes | Immobilon_P, Millipore | IPVH00010 |
| NC membranes | Amersham, Cytiva | 10600001 |

## Plant material and growth condition

Arabidopsis mutants used in this study are all in Col genetic background. The *lrx1* mutant (allele *lrx1-1*) was previously described in (Diet et al, 2006). *pp2c12-2* (SALK 095074C) was obtained from the Arabidopsis stock center. *pp2c12-3*, *pp2c15* (At1g68410) and *pp2ch3* (At1g09160) were generated using CRISPR/Cas9-based vectors. *fer-4*, *fer-5* and *fer-4* lines are described by (Duan et al, 2010), the *bak1* alleles are described by (Schwessinger et al, 2011). All new primers for genotyping are listed in Appendix Table S2.

Unless detailed otherwise, Arabidopsis seeds were surface sterilized and plated on ½ MS supplemented with 2% sucrose, 0.5% w/v MES, MS-vitamin mix, 100 g/L myo-inositol, adjusted to pH 5.7, and solidified with 0.6% gelrite (Duchefa). Seeds were stratified on plates for 2 days at 4 °C before transferring to the growth chamber to grow in a vertical orientation with a 16-h light/ 8-h dark cycle at 22 °C. Root hairs were visualized 5 days after germination. To obtain adult plants, seedlings were transferred to soil and grown in the growth chamber at 22 °C with a 16-h light/8-h dark cycle.

## Recombinant DNA construction

For CRISPR/Cas9-mediated mutagenesis, the *pKI1.1* (Tsutsui and Higashiyama, 2017) or *pAGM55261* (Grützner et al, 2021) plasmids were used for single gRNAs targeting *PCH* genes. Oligos used for CRISPR/Cas9-mutagenesis are listed in Appendix Table S2. For complementation assays, the promoter of *PP2C12* (1.5 kb upstream of the start codon) was amplified with the primers *ROL23PromF* and *ROL23PromR* (Appendix Table S2) cloned into *pGPTV-Kan-GFP* vector (modified from pGPTV-Kan (Becker et al, 1992)) by digesting with SpeI and AscI, resulting in *PP2C12::GFP*. The *PP2C12* genomic coding sequence was then amplified with the primers *ROL23CDSF* and *ROL23CDSR* and cloned by in-fusion cloning into *pGPTV-Kan PP2C12::GFP* with AscI, resulting in *PP2C12:: PP2C12-GFP*. The different *PP2C12::PP2C-GFP* constructs were obtained by introducing the coding sequences of PP2Cs, amplified with the primers in Appendix Table S2, by in-fusion cloning into *PP2C12::GFP* cut with AscI. The *PP2C12:: PP2C12-GFP-NES* and *PP2C12::PP2C12-GFP-nes* constructs were obtained by cutting *PP2C12:: PP2C12-GFP* with BsaI and SacI and ligating with the PCR products of *GFPNESnes_F* and *NES_R* or *nes_R* (Appendix Table S2) from corresponding templates (Kuhn et al, 2011) cut with BsaI and SacI. *BiFC* constructs were generated by gateway cloning. Amplified PCR products were cloned into the *pDONR* vector by BP reaction (Invitrogen), and subsequently transferred to *pDEST* vectors with VenusN and VenusC tags (Gehl et al, 2009) by LR reaction (Invitrogen). For *N. benthamiana* transient expression, lab-generated plasmids *35S::FER-FLAG* and *35S::Iti6b-GFP* were used. Asp to Asn mutations in *35S::PP2C12*$^{dead}$-*GFP* were introduced by PCR-based mutagenesis and cloned into the binary vector *pART27* by NotI (Gleave, 1992). For the in vitro phosphatase assays, His-tagged FER cytosolic domain (CD) and MBP-tagged PP2C12 variants were constructed into the *pET28a(+)* and *pMALc4e* vectors, respectively, using in-fusion cloning. All primers used for recombinant DNA construction are listed in Appendix Table S2.

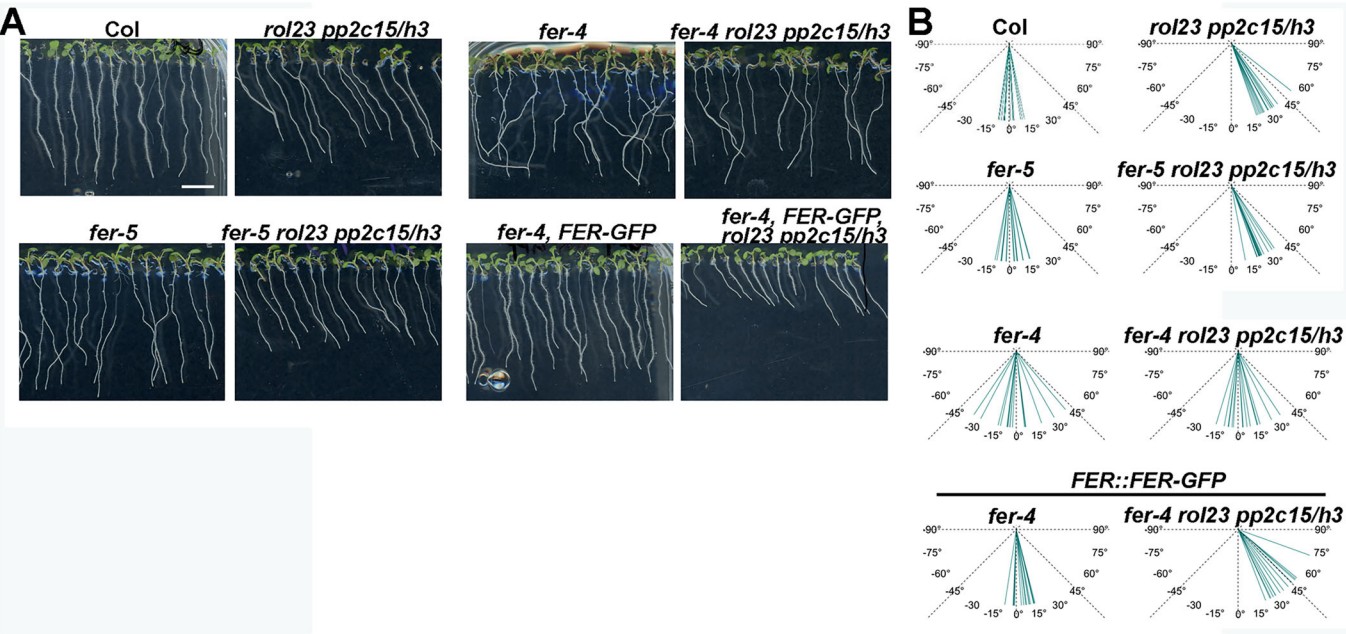

**Figure 9. FER-dependent root skewing in the clade H PP2C triple mutant.**

(A) Images of 7-day-old vertically grown Arabidopsis seedlings are shown of Col, *fer-5*, *fer-4*, *fer-4 FER::FER-GFP* with or without clade H PP2Cs mutated. *rol23 pp2c15 pp2ch3* (labeled *rol23 pp2c15/h3* in (A, B) triple mutant seedlings show root skewing compared to the wild-type Col. This effect is FER-dependent and not seen in the *fer-4 rol23 pp2c15 pp2ch3* quadruple mutant. This effect, however, is observed in the quadruple mutant with the knock-down allele *fer-5* that retains partial FER activity. Scale bar = 5 mm. (B) Quantification of the deviation from the gravitropic vector in samples shown in (A). Fifteen seedlings for each genotype were measured for their root skewing angles. Source data are available online for this figure.

The *FER::FER-GFP* construct is described in (Escobar-Restrepo et al, 2007). The Thr696 variants T696A, T696D, and T696E were introduced by PCR using the primers FER_R4 with either FERT696A_F, FERT696D_F, or FERT696E_F, respectively (Appendix Table S2).

## Plant transformation

Arabidopsis transgenic lines were obtained by standard floral dipping method mediated by Agrobacterium tumefaciens (strain GV3101). For the complementation lines, T1 seeds of plant transformed with *pGPTV-Kan* and *pART27* vectors were selected on Kanamycin selection media. As the CRISPR/Cas9 vector *pKI1.1* and *pAGM55261* carry a FASTRED seed selection marker, transgenic lines were selected based on the RFP fluorescence. DNA from the cauline leaves of the inflorescences of T1 plants were extracted to genotype the plants.

## EMS mutagenesis, CRISPR/Cas9 mutagenesis, and molecular markers

The procedure for EMS mutagenesis of the *lrx1* mutant is described in detail in (Diet et al, 2004). Mutagenized M1 seeds were propagated to the M2 generation in 200 pools of 10 plants each. Two hundred M2 seeds per pool were germinated and grown in a vertical orientation and seedlings with a wild type-like root hair formation were selected for confirmation in the M3 generation. In parallel, *rol* mutant candidates were back-crossed with *lrx1* and the resulting F1 seeds propagated to the F2 generation to analyze segregation of the *rol* mutant phenotype in the F2 generation.

Targeted mutagenesis by CRISPR/Cas9 was performed with vectors described above. Guide RNAs (gRNAs) targeting the individual genes are listed in Appendix Table S2. *PP2C12*, *PP2C15*, and *PP2CH3* were targeted with one gRNA each. Genomic DNA encompassing the gRNA sequences was PCR-amplified of T1 transgenic lines and sequenced using gene-specific primer listed in Appendix Table S2. Plants with mutations were propagated to the T2 generation, where seedlings lacking RFP fluorescence (indicative for absence of CRISPR/Cas9 transgene) and presence of homozygous mutations, determined by sequencing, were selected.

A molecular marker for the *lrx1* mutation is described in (Diet et al, 2004). For *rol23*, genomic DNA was amplified with the primers listed in Appendix Table S2, and the product was digested with StuI, which cuts the *rol23* but not the wild-type variant.

## Whole-genome sequencing

WGS was done as described in (Schaufelberger et al, 2019). Of an F2 population of a backcross of *lrx1 rol23* with *lrx1*, i.e., homozygous for *lrx1* but segregating for *rol23*, 12 seedlings showing a wild type-like phenotype were isolated and the phenotype was confirmed in the F3 generation. 50 seedlings of each of the 12 F3 families were pooled, grinded in liquid nitrogen, and DNA was extracted. DNA sequencing corresponding to a 20-fold coverage was outsourced (BGI Tech Solutions) and obtained for the *lrx1* mutant and the *lrx1 rol23* double mutant pool. Sequences of the *lrx1* and *lrx1 rol23* mutants were separately mapped to the Arabidopsis genome (available on TAIR), and polymorphisms to the *lrx1* sequence were subtracted from those of the *lrx1 rol23* mutant. The resulting list of *rol23*-specific SNPs were filtered

for mutations changing the sequence of encoded proteins based on the annotation provided by the TAIR database. For the SNP-selection, we used in-house Perl scripts which can be provided upon request.

## RALF-induced root growth inhibition assays

Surface-sterilized seeds were vertically grown on ½ MS agar plates as described above, but supplemented with 1% sucrose, for 4 days. Six 4-day-old seedlings were then transferred to each well of a 12-well plate containing 4 mL ½ MS liquid media (as described above but without gelrite) with 2 µM RALF peptide for additional 2 days of growth. The RALF peptides were synthesized by SciLight Biotechnology LLC with a purity of 85% (RALF1: ATTKYISYQSLKRNSVPCSRRGASYYNCQN GAQANPYSRGCSKIARCRS;). In the control group, seedlings were grown under identical conditions in peptide-free media. The seedlings were then laid out on solid medium for scanning. Primary root length was measured and quantified by Fiji (https://imagej.net/Fiji; Schindelin et al, 2012). Twelve seedlings of each genotype under each condition were measured. Three independent experiments were performed.

## Protein expression, extraction, and co-immunoprecipitation

For transient protein expression, fully expanded leaves of 4-week-old *N. benthamiana* were infiltrated with Agrobacteria GV3101 carrying the constructs of interest. Agrobacteria containing the *P19* suppressor vector were co-infiltrated with all the constructs to avoid silencing. Transiently transformed *N. benthamiana* leaves were harvested 48-h post-infiltration and grinded to powder in liquid nitrogen. For FER-PP2C12(DN) interaction analysis, proteins were isolated with 50 mM Tris-HCl pH 7.5, 150 mM NaCl, 10% glycerol, 2 mM EDTA pH8, 5 mM dithiothreitol, 1% IGEPAL CA-630 (Sigma), 1% protease inhibitor (Roche) and 1 mM PMSF. For protein extraction from Arabidopsis transgenic lines, tissue from thirty to fifty 14-day-old seedlings per line grown in liquid ½ MS were used. Protein extraction was performed as above. For immunoprecipitations, the crude protein extract was incubated with GFP-trap (Chromotek) agarose beads at 4 °C for 3–4 h or overnight. Beads were then washed five times. Samples were then boiled with 1x SDS-PAGE loading buffer at 95 °C for 10 min before SDS-PAGE separation.

## Immunoblotting

Protein samples were separated by 10% SDS-PAGE (130 V for 90 min) and blotted onto PVDF membranes (100 V for 90 min), followed by membrane blocking with TBST-5% fat-free milk for one hour. The membrane was probed in the blocking buffer with anti-GFP-HRP (Santa Cruz, 1:5000 dilution) and an anti-FLAG-HRP (Sigma-Aldrich, 1:4000 dilution). Alternatively, an anti-GFP primary antibody (Sigma-Aldrich, 1:5000), followed by an anti-mouse-HRP secondary antibody (Sigma-Aldrich, 1:5000), were used. The endogenous FER was detected by anti-FER ((Xiao et al, 2019), 1:2000 dilution) as primary and anti-rabbit-HRP (Sigma-Aldrich, 1:10,000 dilution) as secondary antibody.

## Recombinant protein expression and purification

*FER* CD was fused to 6xHis in the *pET28a(+)*, *PP2C12* and *PP2C12*[dead] were fused to *MBP* and expressed using *pMAL-c4e*. The

recombinant proteins were expressed in *E. coli* strain BL21(DE3) Rosetta pLysS. Isopropyl b-d-1-thiogalactopyranoside (IPTG, 0.5 mM final concentration) was used to induce protein expression at 16 °C overnight. Bacteria were harvested by centrifugation and resuspended in the lysis buffer containing 40 mM HEPES pH 7.2, 300 mM NaCl, 5% (v/v) glycerol, 2 mM PMSF and 1% protease inhibitor (Roche). The samples were then lysed by sonication and centrifuged to collect the supernatant for the following protein purification. Amylose Resin (NEB) or HisPur Cobalt Resin (Thermo) were used to purify MBP and 6xHis fusions, respectively. Five hundred millimolar imidazole was added to elute proteins bound to His resin, whilst 200 mM maltose was used for the MBP protein elution. 6xHis fusion proteins were further concentrated and subsequently diluted with buffer lacking imidazole to reduce the imidazole concentration to below 1 mM.

## Phosphatase assay

Recombinant PP2C12 and PP2C12[rol23] fused to MBP were purified as described above. To test for protein abundance, proteins were separated by 10% SDS-PAGE, blotted on NC membrane (Amersham) by semi-dry blotting (BioRad), and immune-detected with an α-MBP antibody (NE Biolabs, 1:5000 dilution). The phosphatase assay (Ser/Thr Phosphatase Assay System, Promega) was performed with equal amounts of recombinant protein in 40 mM HEPES pH 7.2, 5 mM MgCl₂, 5 mM MnCl₂, 200 mM NaCl, 5% glycerol, 1 mM DTT, at 30 °C for 90 min following the manufacturer's instructions.

## Phosphoprotein immunoblotting

The total protein extracted from Arabidopsis seedlings were separated by 10% SDS PAGE supplemented with 50 mM Phostag reagent (Wako) and 50 mM MnCl₂, followed by subsequent blotting onto PVDF membrane (100 V for 3 h). Endogenous FER was detected by anti-FER (Xiao et al, 2019), 1:2000 dilution) and anti-rabbit-HRP (Sigma-Aldrich, A0545, 1:10,000 dilution).

One microgram of autophosphorylated FER 6xHis was incubated with 4 µg of MBP tagged PP2C12 variants in the phosphatase buffer (40 mM HEPES pH 7.2, 5 mM MgCl₂, 5 mM MnCl₂, 200 mM NaCl, 5% glycerol, 1 mM DTT) at 25 °C for 5–90 min. Reactions were quenched by boiling the samples with 1x SDS loading buffer at 95 °C for 10 min. Phosphorylation was then monitored by immunoblotting with anti-pThr (Cell Signalling Technology 9381, 1:1000 dilution in TBST-5% gelatin from cold water fish skin).

## LC-MS/MS analysis, phosphopeptide identification, and quantification

Phosphatase reactions performed with recombinant proteins produced in *E. coli* were mixed with an equal volume (25 µl) of reduction/alkylation buffer containing 7.4 M urea, 10 mM tris(2-carboxyethyl)phosphine-HCl, and 50 mM 2-chloroacetamide followed by incubation for 60 min at 60 °C in the dark with shaking at 500 rpm. Samples were then diluted to a final volume of 250 µl with 50 mM ammonium bicarbonate (ABC) and 100 ng of trypsin (10 ng/µl in ABC) was added to digest the proteins. Digests were incubated overnight at 37 °C. The next day, digests were acidified by the addition of 2.5 µl of 10% trifluoroacetic acid. Peptides were

cleaned up using the StageTip method, dried down, redissolved in 20 µl of MS buffer (3% acetonitrile, 0.1% formic acid), and loaded in to MS sample vials.

Liquid chromatography-tandem mass spectrometry (LC-MS/MS) was performed using an M-Class HPLC (Waters) coupled to an Exploris 480 orbitrap mass spectrometer (Thermo Fisher Scientific). Two microliters of each peptide sample were loaded onto a nanoEase M/Z Symmetry C18 100 Å, 5 µm, 180 µm × 20 mm trap column at a flow rate of 15 µl·min$^{-1}$ in Buffer A (0.1% formic acid in water) for 2 min. Peptide separation was performed on a nanoEase M/Z HSS C18 T3 Col 100 Å, 1.8 µm, 75 µm × 250 mm analytical column at a flow rate of 300 nl·min$^{-1}$ using a 110 min gradient from 5% to 95% buffer B (buffer B = 0.1% formic acid in acetonitrile). Peptides were ionized at a spray voltage of 2.4 kV and a capillary temperature of 270 °C. The mass spectrometer was operated in data-dependent mode with 3 s between master scans. Full scan MS spectra (350–1200 *m/z*) were collected with a maximum injection time of 45 ms at a resolution of 120,000 and a normalized AGC target of 300%. Dynamic exclusion was set to 15 s. High-resolution MS2 spectra were collected in the orbitrap with a maximum injection time of 200 ms at 30,000 resolution (isolation window 1.2 *m/z*), a normalized AGC target of 100%, and a normalized collision energy of 30%. Only precursors with charge states from 2 to 7 were selected for fragmentation.

The mass spectrometry proteomics data was handled using the local laboratory information management system (LIMS) (Türker et al, 2010). Raw files data were processed using MSFragger (version 4.1) (Kong et al, 2017) implemented in fragpipe (version 22) using default closed-search settings unless otherwise specified. MS/MS spectra were searched against an *E. coli* database (Uniprot ID UP000000625, downloaded June 4, 2024) with sequences for FER, ROL23, and ROL23$^{dead}$ added. Trypsin was set as protease, allowing for up to 2 missed cleavages. Carbamidomethylation of cysteines was set as a fixed modification with methionine oxidation, N-terminal acetylation, and phosphorylation of serine, threonine and tyrosine set as variable modifications. Label-free quantification was performed IonQuant (version 1.10.27) (Yu et al, 2021) using default settings implemented in fragpipe with the match between runs feature enabled.

### Bimolecular Fluorescence Complementation Assay

Full length *FER*, *PP2C12* in *pDEST-VenusN* and *pDEST-VenusC* were expressed pairwise in *N. benthamiana*, together with the plasma membrane marker (*35S:REM* 1.2 mRFP, Bücherl et al, 2017) and *P19*. The plasma membrane marker Iti6b was used as the negative control. Fourty eight hours post- infiltration, leaf discs were collected to monitor the mVenus signal with Stellaris confocal microscope.

### Confocal microscopy, image processing and data quantification

All samples were imaged using Stellaris inverted confocal microscope with white light lasers. For BiFC assay, the lower epidermis of the leaf discs was immersed in MilliQ water, and mVenus fluorophore was excited at 514 nm and emitted light recorded between 520 and 555 nm. mRFP was exited at 561 nm and the fluorescence emission was collected between 575 and 650 nm.

Optimized identical acquisition parameters were applied to analyze comparative samples. For FER-GFP endocytosis, 5- to 7-day-old Arabidopsis seedlings were immersed in ½ MS liquid media. GFP fluorescence was monitored with 488 nm excitation/500–550 nm emission.

Images were displayed using Leica LAS X software and Fuji, and assembled with Adobe Illustrator. For the endocytosis analysis, particle sizes >0.1 µm$^2$, 0.25 µm$^2$, 0.5 µm$^2$ were used for quantification. For each treatment/transgenic line, five seedlings, four cells from the transition zones per seedling were used for quantification. The number of particles per 100 µm$^2$ was calculated.

For live-cell imaging, plasma membranes of 5- to 7-day-old Arabidopsis seedling were stained in solution containing 2 µM FM4-64 (N-(3-triethylammoniumpropyl)-4-(6-(4-(diethylamino) phenyl) hexatrienyl) pyridinium dibromide). Lines expressing FER-GFP were imaged 30 to 45 min after treatment in ½ MS media with DMSO (mock) or in presence of RALF1 (1 µM). For imaging cell invagination, seedlings expressing PP2C12-GFP were incubated in presence of 1 µM RALF1 for at least 1 h.

### Quantification of root skewing

Seedlings were grown for 6 days in a vertical orientation and the plates were scanned. Deviation of the root apices from the Y axis of growth was used as parameter to quantify skewing as described elsewhere (Gupta et al, 2024).

### Accession number of genes used in this study

ROL23/PP2C12, At1g47380; PP2C15, At1g68410; PP2CCH3, At1g09160; FER, At3g51550; LRX1, At1g12040.

## Data availability

This study includes no data deposited in external repositories.

The source data of this paper are collected in the following database record: biostudies:S-SCDT-10_1038-S44318-025-00614-x.

## Peer review information

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

## Acknowledgements

We thank the infrastructure team at IPMB was continuous support with growth facilities and Matthias Heuberger for support with the phylogenetic analysis of the PP2C family. This work was funded by the University of Zurich, the Swiss National Science Foundation grants No. 31_61419.00, 31003A_166577, and 310030_192495 to CR, and 320030_228294 to CZ, and by the European Research Council (ERC) under the European Union (EU)'s Horizon 2020 research and innovation program under grant agreement No. 773153 (grant "IMMUNO-PEPTALK" to CZ). ADFF was partially supported by an EMBO Postdoctoral Fellowship (ALTF 580-2022).

## Author contributions

**Xiaoyu Hou**: Data curation; Formal analysis; Validation; Investigation; Visualization; Methodology; Writing—original draft; Writing—review and editing. **Kyle W Bender**: Data curation; Formal analysis; Supervision; Investigation; Visualization; Methodology; Writing—original draft; Writing—review and editing. **Álvaro D Fernández-Fernández**: Data curation; Formal analysis; Investigation; Methodology; Writing—original draft; Writing—review and editing. **Gabor Kadler**: Data curation; Formal analysis. **Shibu Gupta**: Data curation; Formal analysis. **Mona Häfliger**: Data curation; Formal analysis. **Amandine Guérin**: Data curation; Formal analysis; Methodology. **Anouck Diet**: Formal analysis. **Stefan Roffler**: Data curation; Formal analysis. **Daniela Campanini**: Formal analysis. **Thomas Wicker**: Data curation; Supervision. **Cyril Zipfel**: Supervision; Writing—review and editing. **Christoph Ringli**: Conceptualization; Data curation; Formal analysis; Supervision; Funding acquisition; Investigation; Visualization; Methodology; Writing—original draft; Project administration; Writing—review and editing.

Source data underlying figure panels in this paper may have individual authorship assigned. Where available, figure panel/source data authorship is listed in the following database record: biostudies:S-SCDT-10_1038-S44318-025-00614-x.

## Disclosure and competing interests statement

The authors declare no competing interests.

