## [Peer Review File · The EMBO Journal]

The Arabidopsis phosphatase PP2C12 negatively regulates LRX-RALF-FER-mediated cell wall integrity sensing

Xiaoyu Hou, Kyle Bender, Álvaro Daniel Fernández-Fernández, Gabor Kadler, Shibu Gupta, Mona Häfliger, Amandine Guerin, Anouck Diet, Stefan Roffler, Daniela Campanini, Thomas Wicker, Cyril Zipfel, and Christoph Ringli

Corresponding author(s): Christoph Ringli (chringli@botinst.uzh.ch)

Review Timeline:

Submission Date:	11th Nov 24
Editorial Decision:	2nd Jan 25
Revision Received:	30th Jul 25
Editorial Decision:	8th Sep 25
Revision Received:	15th Sep 25
Accepted:	6th Oct 25

Editor: William Teale

Transaction Report:

Dear Christoph,

Thank you again for the submission of your manuscript entitled "The phosphatase PP2C12 is a negative player in LRX-RALF-FER-mediated cell wall integrity sensing" and for your patience during the review process. We have now received the reports from the referees, which I copy below.

As you can see from their comments, all referees considered this detailed study to be both timely and well executed. Referee #3 suggests an extensive list of extra experiments; I suggest that we discuss this list over Zoom and together decide on a list of priorities for your attention. Some elucidation of specificity and the nature of key genetic interactions will be necessary before your manuscript can be published in The EMBO Journal.

Based on the overall interest expressed in the reports, I would like to invite you to address the comments of all referees in a revised version of the manuscript. I should add that it is The EMBO Journal policy to allow only a single major round of revision and that it is therefore important to resolve the main concerns at this stage. I believe the concerns of the referees are reasonable and addressable, but please contact me if you have any questions, need further input on the referee comments or if you anticipate any problems in addressing any of their points. Please also suggest some convenient times for a Zoom call. I include instructions below for preparing your manuscript for resubmission.

I would also like to point out that as a matter of policy, competing manuscripts published during this period will not be taken into consideration in our assessment of the novelty presented by your study ("scooping" protection). We have extended this 'scooping protection policy' beyond the usual 3 month revision timeline to cover the period required for a full revision to address the essential experimental issues. Please contact me if you see a paper with related content published elsewhere to discuss the appropriate course of action.

Again, please contact me at any time during revision if you need any help or have further questions.

Thank you very much again for the opportunity to consider your work for publication. I look forward to your revision.

Best regards,

William

William Teale, Ph.D.
Editor
The EMBO Journal

When submitting your revised manuscript, please carefully review the instructions below and include the following items:

- 1) a .docx formatted version of the manuscript text (including legends for main figures, EV figures and tables). Please make sure that the changes are highlighted to be clearly visible.
- 2) individual production quality figure files as .eps, .tif, .jpg (one file per figure).
- 3) a .docx formatted letter INCLUDING the reviewers' reports and your detailed point-by-point response to their comments. As part of the EMBO Press transparent editorial process, the point-by-point response is part of the Review Process File (RPF), which will be published alongside your paper.
- 4) a complete author checklist, which you can download from our author guidelines ([https://wol-prod-cdn.literatumonline.com/pb-assets/embo-site/Author Checklist%20-%20EMBO%20J-1561436015657.xlsx](https://wol-prod-cdn.literatumonline.com/pb-assets/embo-site/Author%20Checklist%20-%20EMBO%20J-1561436015657.xlsx)). Please insert information in the checklist that is also reflected in the manuscript. The completed author checklist will also be part of the RPF.
- 5) Please note that all corresponding authors are required to supply an ORCID ID for their name upon submission of a revised manuscript.
- 6) We require a 'Data Availability' section after the Materials and Methods. Before submitting your revision, primary datasets produced in this study need to be deposited in an appropriate public database, and the accession numbers and database listed

under 'Data Availability'. Please remember to provide a reviewer password if the datasets are not yet public (see <https://www.embopress.org/page/journal/14602075/authorguide#datadeposition>). If no data deposition in external databases is needed for this paper, please then state in this section: This study includes no data deposited in external repositories. Note that the Data Availability Section is restricted to new primary data that are part of this study.

Note - All links should resolve to a page where the data can be accessed.

8) For data quantification: please specify the name of the statistical test used to generate error bars and P values, the number (n) of independent experiments (specify technical or biological replicates) underlying each data point and the test used to calculate p-values in each figure legend. The figure legends should contain a basic description of n, P and the test applied. Graphs must include a description of the bars and the error bars (s.d., s.e.m.).

9) We would also encourage you to include the source data for figure panels that show essential data. Numerical data can be provided as individual .xls or .csv files (including a tab describing the data). For 'blots' or microscopy, uncropped images should be submitted (using a zip archive or a single pdf per main figure if multiple images need to be supplied for one panel). Additional information on source data and instruction on how to label the files are available at .

10) We replaced Supplementary Information with Expanded View (EV) Figures and Tables that are collapsible/expandable online (see examples in <https://www.embopress.org/doi/10.15252/embj.201695874>). A maximum of 5 EV Figures can be typeset. EV Figures should be cited as 'Figure EV1, Figure EV2" etc. in the text and their respective legends should be included in the main text after the legends of regular figures.

12) Our journal encourages inclusion of *data citations in the reference list* to directly cite datasets that were re-used and obtained from public databases. Data citations in the article text are distinct from normal bibliographical citations and should directly link to the database records from which the data can be accessed. In the main text, data citations are formatted as follows: "Data ref: Smith et al, 2001" or "Data ref: NCBI Sequence Read Archive PRJNA342805, 2017". In the Reference list, data citations must be labeled with "[DATASET]". A data reference must provide the database name, accession number/identifiers and a resolvable link to the landing page from which the data can be accessed at the end of the reference. Further instructions are available at .

13) In order to increase the reproducibility and reach of your work, The EMBO Journal includes a table of reagents that were used in the study. Please provide this along with your revisions.

Further instructions for preparing your revised manuscript:

We realize that it is difficult to revise to a specific deadline. In the interest of protecting the conceptual advance provided by the work, we recommend a revision within 3 months (2nd Apr 2025). Please discuss the revision progress ahead of this time with the editor if you require more time to complete the revisions. Use the link below to submit your revision:

Referee #1:

The authors present a comprehensive study identifying PP2C12, a clade H PP2C, as a key regulator of the LRX1 pathway. They demonstrate that PP2C12 specifically suppresses the short root hair phenotype of the *lrx1* mutant, and suggest a unique role for clade H PP2Cs in this pathway. Indeed, the other two members of this clade, but not PP2C members from three different clades (D, E, L), can partially complement the *lrx1 pp2c12* mutant.

The authors propose that PP2C12 functions downstream of the RALF-FERONIA signaling module. They provide strong evidence for this model, including the partial complementation of a weak *fer* allele by *pp2c12*, and the direct interaction between PP2C12 and the FERONIA kinase domain. A key finding is that PP2C12 directly dephosphorylates a threonine residue in FERONIA's activation loop, acting as a negative regulator of its kinase activity. This regulation is crucial for the regulation of FERONIA activity, as evidenced by the constitutive endocytosis of FERONIA in a clade H PP2C triple mutant (endocytosis being a whole mark of FER activation by RALF or pectins). Additionally, clade H PP2Cs influence root skewing, likely through FER-dependent microtubule regulation.

Overall, this is a well-executed study that significantly advances our understanding of the LRX1-FERONIA signaling pathway. It shows that direct dephosphorylation of receptor kinases by cytosolic phosphatases is a widespread regulation. The finding that clade H PP2C but not PP2Cs from 3 other clades can specifically suppress the *lrx1* phenotype is very interesting. Addressing the following points could further enhance the strength of the conclusions.

Comments and Suggestions:

1. PP2C12 Overexpression Phenotype: The quantification data for PP2C12-GFP overexpression in Fig. 3D is missing. It would be valuable to clarify the root hair phenotype of these overexpression lines and whether they can phenocopy a *fer* phenotype.
2. Hypersensitivity to RALF1: Given the predicted reduced inhibition of FERONIA in the *rol23 pp2c15 pp2ch3* triple mutant, it would be interesting to test its response to RALF1 at lower concentrations to potentially reveal a hypersensitive phenotype.
3. Endocytosis regulation in clade H PP2Cs mutant: RALF-activated FERONIA triggers its own endocytosis but also non-selective endocytosis of many proteins. Is the effect of the *rol23 pp2c15 pp2ch3* triple mutant specific to FER endocytosis (caused by its specific phosphostatus) or is it a general increase in endocytic flux (triggered by FER activation)? This can be

tested by using the endocytic tracer FM4-64 without the need to cross new membrane markers in the triple mutant.

4. Specificity of PP2C-FERONIA Interaction: Including a PP2C from another clade as a control in BIFC and co-IP experiments would strengthen the specificity of the interaction between clade H PP2Cs and FERONIA, particularly for the BIFC experiment, which is prone to false positives.

Minors:

5. Microtubule Array Images: The low resolution of the microtubule array images limits their interpretability. Providing higher-resolution images would be important. Alternatively, the authors could consider omitting this data, since this is indirect evidence (the angle of the microtubules could be either a cause or a consequence of root skewing)

6. PP2C12 DN Nomenclature: To avoid confusion, I recommended using "PP2C12dead" instead of "PP2C12 DN" to denote the dead version, as the Asp (D) to Asn (N) mutant does not exhibit a dominant-negative (DN) effect.

Referee #2:

The submitted manuscript by Hou et al. is concerned with the role of PP2C12, a type 2C phosphatase, in the LRX-RALF-FERONIA signalling pathway. The authors identified PP2C12 as a repressor of *lrx1_23* (*rol23*) genes in the EMS screen for *lrx1* suppressors. PP2C12 was shown to interact with FERONIA (FER) and dephosphorylate the FER activation loop, supposedly functioning upstream of the RALF1-FERONIA signalling module. The manuscript is timely, very well-written, and will be of interest to researchers in the plant molecular and developmental biology field. However, I have several concerns regarding the experiments and their interpretation described in the manuscript.

Major concerns

1)The authors claim that clade H PP2C isoforms have an impact on FER endocytosis (Figure However, the changes in FER localization could be caused by altering other vesicle trafficking pathways (e.g. exocytosis). How would the localization pattern of FER change upon the CME inhibition or addition of RALF?

2)The authors described that PP2C12 is present in the cytoplasm and the nucleus. However, dephosphorylation of FER supposedly happens at the plasma membrane. The authors should discuss this discrepancy. How does the PP2C12 localization pattern change after the RALF treatment?

3)The authors describe that PP2C12 dephosphorylates the FER cytoplasmic domain (CD) in vitro (figure 6A). The authors might comment on how the phosphorylated FER CD was obtained. Is the FER CD autophosphorylated in *E.coli*? Based on the provided CBB gel, assessing the purity of recombinantly expressed proteins is difficult. The authors should provide anti-HIS and anti-MBP western blots.

4)The authors state that PP2C12 has an inhibitory activity on FER. However, this is not shown. Are apoplastic alkalization and Ca²⁺ fluxes affected by PP2C12?

Minor concerns

1)The authors note that the interaction between phosphatases and their substrates is often transient. Therefore, they used a dominant negative version of PP2C12 in co-immunoprecipitation assays. However, they used WT in the BIFC assay. The authors might want to discuss a different selection for these two interaction assays.

2)The manuscript does not reference the FER activation loop.

3)What does the unlabelled plant (third from the left) in Figure 2A represent?

4)The authors state (page 10): "Interestingly, PP2C12-GFP overexpression lines showed significantly reduced sensitivity to RALF1 (Fig. 3D, Appendix Fig. S3B,C,D)". I could not find this in Fig. 3D.

5)The phylogenetic analysis in Figure S1 is not described in the Materials and Methods section.

Referee #3:

This study highlights the critical role of the LRX1-RALF-FER signaling pathway in regulating cell wall integrity (CWI) in Arabidopsis. The LRX proteins are identified as key mediators linking cell wall dynamics to intracellular signaling by forming a molecular platform with RALF peptides and the receptor-like kinase FERONIA (FER). Specifically, LRX1, which is predominantly expressed in root hairs, is essential for their proper development, as evidenced by defective root hair phenotypes in *lrx1* mutants. Furthermore, the identification of *rol23*, encoding the phosphatase PP2C12, as a suppressor of the *lrx1* mutant

phenotype, provides significant insights into the regulation of this signaling pathway. PP2C12 interacts with FER and dephosphorylates a critical threonine residue (Thr696) in the activation loop of FER. These findings suggest that LRX1 functions upstream of the RALF1-FER signaling module, while PP2C12 modulates FER activity to fine-tune CWI signaling. This offers a new layer of understanding regarding root hair development and the regulation of CWI.

While the study presents intriguing findings, there are several critical concerns that should be addressed before considering the manuscript for publication. My detailed comments are as follows:

Comments:

1. Generation and Analysis of FER-T696A/D Mutants

It is essential to generate FER-T696A and FER-T696D point mutation lines to confirm whether the loss of FER phosphorylation at Thr696 (T696A) results in defective root hair phenotypes, while the phospho-mimetic mutation (T696D) promotes root hair development. This will directly address whether PP2C12's phosphatase activity, which inhibits FER kinase activity, is the primary cause of the root hair defects observed in the *lrx1* mutant.

2. Validation of PP2C12-G66R Phosphatase Activity

Perform phosphatase activity assays to verify whether the *rol23* (PP2C12-G66R) mutant protein has lost or significantly reduced phosphatase activity. This is critical to establish the functionality of the suppressor mutation and its impact on FER-mediated signaling.

3. Interaction of Other PP2CHs Family Members with FER

Confirm whether other members of the PP2CHs subfamily interact with FER and inhibit its kinase activity. This will help determine whether the regulation of FER by PP2C12 is a unique feature or part of a broader regulatory mechanism involving other PP2CHs.

4. Expression Analysis of PP2C Family Members in Root Hairs

Examine GFP fluorescence signals of PP2C12-GFP, PP2C15-GFP, and other PP2Cs family members (e.g., PP2C35, PP2C38, PP2C52, and PP2CH3) in root hairs. This will confirm whether these phosphatases are co-expressed in root hairs and potentially involved in the regulation of FER signaling.

5. Improved Design of In Vitro Kinase Assay (Fig. 6A)

The current design of the in vitro phosphorylation assay (Fig. 6A) is insufficiently rigorous. A more detailed experimental design is needed to clearly demonstrate the effect of PP2C12 on FER-CD dephosphorylation. The existing results do not convincingly reflect the significant impact of PP2C12 on FER kinase activity.

6. Correction of Immunoblotting Terminology (Fig. 3A/B)

In Fig. 3A/B, the term "IB: GFP-Trap" should be revised to "IP: GFP-Trap" to accurately reflect the experimental procedure.

7. Consistency in Input Controls (Fig. 5A/B)

In Fig. 5A, the input is visualized using Ponceau S, while in Fig. 5B, Coomassie Brilliant Blue (CBB) is used. This inconsistency in experimental design should be addressed. Furthermore, the anti-FER signal in Fig. 5B appears unreliable and needs to be validated with more robust experimental data.

8. Additional Evidence for Direct Interaction (GST Pull-down Assay)

Include results from GST pull-down assays to confirm the direct interaction between FER-CD and PP2C12, as well as other homologous PP2CHs members. This additional evidence will strengthen the claim of direct interaction between these proteins.

Responses to reviewers' comments

We are grateful for the detailed comments of the reviewers on our manuscript and time they spent for suggestions on how to improve our work. We have addressed all the points raised, either by providing new experiments, new descriptions, or justification, why the desired information could not be obtained or would be suboptimal.

All reviewers comments are in **bold**. All our responses non-bold.

Fig. 7 represents an entirely new set of data, Fig. 3, 4, and 6 have new panels, and Fig. 8 (now Fig. 9) C,D was removed (as suggested by one reviewer). Appendix Fig. S6 is new and Fig. S1, S3, and S4 contain new panels.

For every point, we give the Figure that was changed or added, and the page number where the text was modified. The page number refers to the word file of the manuscript simple markup track mode. (The PDF conversion in the submission system slightly shifts the lanes, which is why the numbers given below might not always be correct in the PDF.)

Response to Referee #1:

The authors present a comprehensive study identifying PP2C12, a clade H PP2C, as a key regulator of the LRX1 pathway. They demonstrate that PP2C12 specifically suppresses the short root hair phenotype of the *lrx1* mutant, and suggest a unique role for clade H PP2Cs in this pathway. Indeed, the other two members of this clade, but not PP2C members from three different clades (D, E, L), can partially complement the *lrx1 pp2c12* mutant.

The authors propose that PP2C12 functions downstream of the RALF-FERONIA signaling module. They provide strong evidence for this model, including the partial complementation of a weak *fer* allele by *pp2c12*, and the direct interaction between PP2C12 and the FERONIA kinase domain. A key finding is that PP2C12 directly dephosphorylates a threonine residue in FERONIA's activation loop, acting as a negative regulator of its kinase activity. This regulation is crucial for the regulation of FERONIA activity, as evidenced by the constitutive endocytosis of FERONIA in a clade H PP2C triple mutant (endocytosis being a whole mark of FER activation by RALF or pectins). Additionally, clade H PP2Cs influence root skewing, likely through FER-dependent microtubule regulation.

Overall, this is a well-executed study that significantly advances our understanding of the LRX1-FERONIA signaling pathway. It shows that direct dephosphorylation of receptor kinases by cytosolic phosphatases is a widespread regulation. The finding that clade H PP2C but not PP2Cs from 3 other clades can specifically suppress the *lrx1* phenotype is very interesting. Addressing the following points could further enhance the strength of the conclusions.

Comments and Suggestions:

1. PP2C12 Overexpression Phenotype: The quantification data for PP2C12-GFP overexpression in Fig. 3D is missing. It would be valuable to clarify the root hair phenotype of these overexpression lines and whether they can phenocopy a *fer* phenotype.

The root hair phenotype assessment and quantification are described (page 7, lanes 22-25) and shown in Appendix Fig. S3E-G. Root hairs frequently form bulges at the basis but elongate comparable to the wild-type Col. The discrepancy to overexpression in the *lrx1 rol23* background (Fig. 2) is interesting and discussed (page 12, lanes 19-23).

2. Hypersensitivity to RALF1: Given the predicted reduced inhibition of FERONIA in the *rol23 pp2c15 pp2ch3* triple mutant, it would be interesting to test its response to RALF1 at lower concentrations to potentially reveal a hypersensitive phenotype.

This is indeed an interesting suggestion. The experiment was conducted repeatedly, but results obtained were not consistent. Sometimes, the triple mutant would be hypersensitive to RALF1, sometimes not. Therefore, we prefer not to make a statement on that point.

3. Endocytosis regulation in clade H PP2Cs mutant: RALF-activated FERONIA triggers its own endocytosis but also non-selective endocytosis of many proteins. Is the effect of the *rol23 pp2c15 pp2ch3* triple mutant specific to FER endocytosis (caused by its specific phosphostatus) or is it a general increase in endocytic flux (triggered by FER activation)? This can be tested by using the endocytic tracer FM4-64 without the need to cross new membrane markers in the triple mutant.

We have done this analysis, and FM4-64 does not show increased labelling in the *rol23 pp2c15 pp2ch3* triple mutant line. Hence, based on these results, the effect seems specific for FER-GFP. This is shown in Appendix Fig. S6A and described on page 10, lanes 18-20.

In the same set of experiments, we also analyzed whether the RALF1-induced endocytosis of FER-GFP is altered in the *rol23 pp2c15 pp2ch3* triple mutant background. Here, no difference was observed (Appendix Fig. S6A, and page 10, last paragraph).

The Materials & Methods section was complemented accordingly (page 21, first paragraph).

These experiments are discussed on page 14, first paragraph.

4. Specificity of PP2C-FERONIA Interaction: Including a PP2C from another clade as a control in BIFC and co-IP experiments would strengthen the specificity of the interaction between clade H PP2Cs and FERONIA, particularly for the BIFC experiment, which is prone to false positives.

Other members of the PP2C family have been demonstrated to interact with FER. Hence, the potential interaction of any PP2C with FER does neither prove the interaction of PP2C12 wrong, nor does it provide evidence that any PP2C-FER interaction has the same biological effect as PP2C12. Using any PP2C would actually not be a better negative control. Different approaches have been taken to show interaction of PP2C12 and FER kinase domain, and with the new data on the phospho-dead FER^{T696A} version (Fig. 7, page 9-10), we think we have a strong point of PP2C12 being a regulator of the FER kinase domain.

Minors:

5. Microtubule Array Images: The low resolution of the microtubule array images limits their interpretability. Providing higher-resolution images would be important. Alternatively, the authors could consider omitting this data, since this is indirect evidence (the angle of the microtubules could be either a cause or a consequence of root skewing)

Due to the new data produced that expand this manuscript, we have decided to follow the recommendation of this reviewer and remove this dataset from the manuscript (Fig. 9C,D; corresponding to Fig. 8C,D in the initial submission).

6. PP2C12 DN Nomenclature: To avoid confusion, I recommended using "PP2C12dead" instead of "PP2C12 DN" to denote the dead version, as the Asp (D) to Asn (N) mutant does not exhibit a dominant-negative (DN) effect.

The text and the figure labelling have been changed accordingly throughout the manuscript.

Response to Referee #2:

The submitted manuscript by Hou et al. is concerned with the role of PP2C12, a type 2C phosphatase, in the LRX-RALF-FERONIA signalling pathway. The authors identified PP2C12 as a repressor of *lrx1_23 (rol23)* genes in the EMS screen for *lrx1* suppressors. PP2C12 was shown to interact with FERONIA (FER) and dephosphorylate the FER activation loop, supposedly functioning upstream of the RALF1-FERONIA signalling module. The manuscript is timely, very well-written, and will be of interest to researchers in the plant molecular and developmental biology field. However, I have several concerns regarding the experiments and their interpretation described in the manuscript.

Major concerns

1)The authors claim that clade H PP2C isoforms have an impact on FER endocytosis (Figure However, the changes in FER localization could be caused by altering other vesicle trafficking pathways (e.g. exocytosis). How would the localization pattern of FER change upon the CME inhibition or addition of RALF?

We have analyzed RALF1-induced endocytosis in more detail. RALF1 induces FER-GFP endocytosis, confirming previous findings (Rössling et al, 2024, referenced in the text). The steady-state level of FER-GFP endocytosis in the *pp2ch* triple mutant background is not further increased by RALF1 (Appendix Fig. S6A, and page 10, last paragraph).

As also requested also by Reviewer 1, we used the endocytic tracer FM4-64 and found that the increased endocytosis in the *pp2ch* triple mutant background appears to be specific for FER-GFP. This is shown in Appendix Fig. S6A, and described on page 10, last paragraph. The Materials & Methods section was complemented accordingly (page 21, first paragraph).

We have also looked at treatments with BrefeldinA, an inhibitor of vesicle trafficking. We have not observed any changes in BrefeldinA-induced effects due to the *pp2ch* triple mutations. Considering the complexity of the interpretation, also because the effect of BrefeldinA is not clearly defined, we decided not to mention these non-conclusive experiments.

2)The authors described that PP2C12 is present in the cytoplasm and the nucleus. However, dephosphorylation of FER supposedly happens at the plasma membrane.

The authors should discuss this discrepancy. How does the PP2C12 localization pattern change after the RALF treatment?

The BiFC experiment shown in Appendix Fig. S4B shows that the FER-PP2C12 interaction overlaps with the plasma membrane marker REM1.2-mRFP. Thus, PP2C12 is a cytoplasmic protein but can localize at the plasma membrane. We also show a virtual section through Arabidopsis root cells done by confocal microscopy which revealed co-localization of PP2C12-GFP with the signal of FM4-64 at the plasma membrane (Appendix Fig S4C, last two lanes page 8 – page 9 first lane). A recent publication in Nature Plants demonstrates the role of PP2C12 in regulating a membrane-localized aluminium receptor (Xu et al, referenced in the text), confirming that at least a fraction of PP2C12 localizes to the plasma membrane. Certainly, PP2C12 is not exclusively localizing to the plasma membrane, since one of its target proteins is the transcriptional regulator ABI4 (Bai et al., referenced in the text). This point is referenced on page 6, second paragraph and discussed on page 13, middle paragraph.

PP2C12-GFP localization was analyzed upon RALF1 treatment. Apart from the formation of invaginations also reported by others (Rössling et al, 2024, referenced in the text), we have not found significant re-localization of PP2C12-GFP. This is shown (Appendix Fig. S6B) and described (page 10, last two lanes – page 11 first two lanes).

3)The authors describe that PP2C12 dephosphorylates the FER cytoplasmic domain (CD) in vitro (figure 6A). The authors might comment on how the phosphorylated FER CD was obtained. Is the FER CD autophosphorylated in E.coli?

It is now stated that FER is auto-phosphorylated in bacteria (page 9, lane 5).

Based on the provided CBB gel, assessing the purity of recombinantly expressed proteins is difficult. The authors should provide anti-HIS and anti-MBP western blots.

The requested immunoblot was not performed at the time, since it would not give information on how pure the FER recombinant protein was. The CBB-stained gel shows that recombinant FER was not significantly degraded over time, allowing to compare the phosphoproteomic data. This also confirmed that the overall abundance of FER peptides was consistent in the samples, again excluding major deviations due to instability of FER. We feel that, together, this builds a trustworthy basis for the conducted comparison of the

phosphopeptides. The new data shown in Fig. 7 also confirm the importance of Thr696 identified in the phosphoproteomic analysis and support our conclusions.

4)The authors state that PP2C12 has an inhibitory activity on FER. However, this is not shown. Are apoplastic alkalinization and Ca²⁺ fluxes affected by PP2C12?

There are several parameters and developmental processes (RALF1 sensitivity, root skewing, root hair formation) that were used to demonstrate that PP2C12 acts as a negative regulator of FER-influenced activity. The publication by Kong et al, 2023 demonstrates that phosphorylation of Thr696 influences FER activity (mentioned and referenced in the manuscript), and we show that PP2C12 modifies Thr696 phosphorylation. We feel that our conclusion is justified. We agree with this reviewer that there would be numerous other processes that could be looked at. For practical reasons, we limited our analysis to those that we show. Looking at calcium dynamics is most certainly an interesting aspect, but lies outside the scope of the present study.

Minor concerns

1)The authors note that the interaction between phosphatases and their substrates is often transient. Therefore, they used a dominant negative version of PP2C12 in co-immunoprecipitation assays. However, they used WT in the BIFC assay. The authors might want to discuss a different selection for these two interaction assays.

The data are now shown in slightly altered sequence, with the Arabidopsis Co-IP assay being shown first, to make clear that we started off with the wild type PP2C12. The rather weak signal let us use the phosphatase-dead version of PP2C12 in *N. benthamiana*. Thus, panels A and B in Fig. 5 are swapped.

Indeed, wild type PP2C12 gives a rather weak interaction in Arabidopsis but a strong signal in the BiFC experiment. A possible explanation for this discrepancy is the long incubation times necessary for the Co-IP experiments that allow PP2C12 to partly dephosphorylate and dissociate from FER. This is mentioned in the text, page 8, line 30-33.

2)The manuscript does not reference the FER activation loop.

This information has been included. We cite Nolen et al., 2004, on page 9, line 16, and page 13, line 4.

3)What does the unlabelled plant (third from the left) in Figure 2A represent?

This labelling (*lrx1 rol23*) has been added.

4)The authors state (page 10): "Interestingly, PP2C12-GFP overexpression lines showed significantly reduced sensitivity to RALF1 (Fig. 3D, Appendix Fig. S3B,C,D)". I could not find this in Fig. 3D.

This mistake has been corrected. Fig. 3C is the correct panel.

5)The phylogenetic analysis in Figure S1 is not described in the Materials and Methods section.

This information is now added in the legend of Appendix Fig. S1.

Response to Referee #3:

This study highlights the critical role of the LRX1-RALF-FER signaling pathway in regulating cell wall integrity (CWI) in Arabidopsis. The LRX proteins are identified as key mediators linking cell wall dynamics to intracellular signaling by forming a molecular platform with RALF peptides and the receptor-like kinase FERONIA (FER). Specifically, LRX1, which is predominantly expressed in root hairs, is essential for their proper development, as evidenced by defective root hair phenotypes in *lrx1* mutants. Furthermore, the identification of *rol23*, encoding the phosphatase PP2C12, as a suppressor of the *lrx1* mutant phenotype, provides significant insights into the regulation of this signaling pathway. PP2C12 interacts with FER and dephosphorylates a critical threonine residue (Thr696) in the activation loop of FER. These findings suggest that LRX1 functions upstream of the RALF1-FER signaling module, while PP2C12 modulates FER activity to fine-tune CWI signaling. This offers a new layer of understanding regarding root hair development and the regulation of CWI.

While the study presents intriguing findings, there are several critical concerns that should be addressed before considering the manuscript for publication. My detailed comments are as follows:

Comments:

1. Generation and Analysis of FER-T696A/D Mutants

It is essential to generate FER-T696A and FER-T696D point mutation lines to confirm whether the loss of FER phosphorylation at Thr696 (T696A) results in defective root hair phenotypes, while the phospho-mimetic mutation (T696D) promotes root hair development. This will directly address whether PP2C12's phosphatase activity, which inhibits FER kinase activity, is the primary cause of the root hair defects observed in the *lrx1* mutant.

The analysis of these transgenic lines is now described on page 9/10 and shown in the new Fig. 7. This dataset demonstrates that Thr696 is important for a *pp2c12* mutation having an effect on root hair development. *pp2c12* in plants expressing FER^{T696A} fails to suppress the *lrx1* phenotype, which supports our hypothesis that increased phosphorylation of Thr696 due to the absence of PP2C12 influences root hair development. This is also discussed (page 12, second paragraph). Unfortunately, the phospho-mimetics FER^{T696D} and FER^{T696E} do not mimic the effect of pT696. It is well-known that phospho-mimetics can, but by no means always do, have the effect of phosphorylation of the corresponding amino acid. Thus, this lack of effect is a negative result that does not allow to draw conclusions on the importance of pT696 in suppressing *lrx1*.

2. Validation of PP2C12-G66R Phosphatase Activity

Perform phosphatase activity assays to verify whether the *rol23* (PP2C12-G66R) mutant protein has lost or significantly reduced phosphatase activity. This is critical to establish the functionality of the suppressor mutation and its impact on FER-mediated signaling.

We have investigated this point with a phosphatase activity measurement on recombinant PP2C12 variants. Fig. 4F shows that PP2C12^{*rol23*} indeed has strongly reduced phosphatase activity compared to PP2C12, supporting the view that reduced phosphatase activity causes the *rol23* phenotype. This finding is described, starting on page 8, lane 4, and a brief explanation is provided, why Gly66Asp strongly reduces the phosphatase activity. The “Materials and Methods” section also contains an additional paragraph with the relevant information on the phosphatase assay (starting on page 18, last paragraph).

3. Interaction of Other PP2CHs Family Members with FER

Confirm whether other members of the PP2CHs subfamily interact with FER and inhibit its kinase activity. This will help determine whether the regulation of FER by

PP2C12 is a unique feature or part of a broader regulatory mechanism involving other PP2CHs.

This manuscript has a focus on PP2C12, and we did not have the resources to look into this interesting aspect any further. Clearly, these PP2CH phosphatases do not only work on one substrate protein; reports are published on PP2C12 also interacting with ABI4 and PSKR1/ALR1 and PP2C15 interacting with BAK1, suggesting an interwoven network involving many proteins (discussed on page 13, second paragraph). We believe that complementation of *rol23* by the other PP2CHs provides better evidence for functional redundancy than interaction with FER, since PP2Cs might interact with FER but have other substrate residues and thus different effects on FER activity (such as e.g. ABI2, Chen et al PNAS).

4. Expression Analysis of PP2C Family Members in Root Hairs

Examine GFP fluorescence signals of PP2C12-GFP, PP2C15-GFP, and other PP2Cs family members (e.g., PP2C35, PP2C38, PP2C52, and PP2CH3) in root hairs. This will confirm whether these phosphatases are co-expressed in root hairs and potentially involved in the regulation of FER signaling.

The analysis of publicly available single-cell RNA sequencing data show that all three *PP2CH* members are expressed in root hairs, albeit at different levels. This is now shown in Appendix Fig. S1D and described in the text (page 5, lanes 27-29).

Importantly, for the complementation experiments, all the *PP2Cs* were expressed under the *PP2C12* promoter to confer expression in the same tissues including root hairs. It is possible that other, non-H clade, *PP2Cs* have the same function as *PP2C12* but in different tissue. Alternatively, they can be expressed in the same tissue but have a different function. Hence, the expression profiles alone of the *PP2Cs* do not provide evidence for their functional specificity, and expression under the same promoter is necessary to provide this information, which is shown in Fig. 2. The expression profiles of the individual non-H clade *PP2C* was not an objective of our analysis.

5. Improved Design of In Vitro Kinase Assay (Fig. 6A)

The current design of the in vitro phosphorylation assay (Fig. 6A) is insufficiently rigorous. A more detailed experimental design is needed to clearly demonstrate the effect of PP2C12 on FER-CD dephosphorylation. The existing results do not convincingly reflect the significant impact of PP2C12 on FER kinase activity.

Please note that we did not perform *in vitro* phosphorylation assays. We performed two experiments to measure the *in vitro* dephosphorylation of the FER kinase domain by PP2C12. In the first experiment, we incubated FER-CD and PP2C12 (phosphatase dead or WT) for different times and measured dephosphorylation by phospho-antibody immunoblotting. In this experiment, we observed a modest decrease in immunoblot signal with the WT phosphatase. In the second experiment, we measured dephosphorylation of FER-CD by PP2C12 by LC-MS/MS to localize the sites targeted by the phosphatase. In this experiment, we observed the clear dephosphorylation of Thr696 – a phosphosite required for full activation of FER (Kong et al., 2023, referenced in the text) – by WT but not phosphatase-inactive PP2C12. As this site lies within the FER activation loop, we conclude that PP2C12 might directly regulate FER activity through activation loop dephosphorylation. We think, and hope that the reviewer will agree, that the inclusion of phosphatase-dead versions of PP2C12 in both of these assays serves as a sufficient control. P-values and the distinction between double and triply charged precursor ions (represented by circles and triangles, respectively) in the Fig. 6B, right panel, were added to improve the information content.

6. Correction of Immunoblotting Terminology (Fig. 3A/B)

In Fig. 3A/B, the term "IB: GFP-Trap" should be revised to "IP: GFP-Trap" to accurately reflect the experimental procedure.

This mistake, which is in Fig. 5A/B was corrected. Also, in Fig. 4C, "IP: α -GFP" was corrected to " α -GFP" (since no immuno-precipitation was performed).

7. Consistency in Input Controls (Fig. 5A/B)

In Fig. 5A, the input is visualized using Ponceau S, while in Fig. 5B, Coomassie Brilliant Blue (CBB) is used. This inconsistency in experimental design should be addressed.

Ponceau S and CBB staining visualize equal loading of the gels and do so equally well. Ponceau S was used in the experiments with *Arabidopsis*, CBB with *N. benthamiana*. We apologize for not having the resources and personnel to repeat these complex experiments to address this minor point.

Furthermore, the anti-FER signal in Fig. 5B appears unreliable and needs to be validated with more robust experimental data.

We would also have preferred to have a stronger FER signal. However, even with several attempts, the signal did not get stronger, which probably reflects the transient nature of the PP2C12-FER interaction, but is not a sign for lack of reliability. To support this finding in Arabidopsis, we have done the Co-IP and split-BiFC experiments in tobacco.

8. Additional Evidence for Direct Interaction (GST Pull-down Assay)

Include results from GST pull-down assays to confirm the direct interaction between FER-CD and PP2C12, as well as other homologous PP2CHs members. This additional evidence will strengthen the claim of direct interaction between these proteins.

Co-IP experiments with *E. coli*-expressed recombinant proteins did not show an interaction between the FER kinase domain and PP2C12. This might be because of necessary posttranslational modifications that are lacking in bacterial-expressed proteins, or additional proteins that are involved but missing in this setup that would stabilize the PP2C12-FER complex. For this reason, we were also not able to compare binding of other PP2Cs to FER. Importantly, as mentioned in our response to Reviewer 1, other PP2Cs might bind FER but have a different biological function, i.e. different phosphate groups as substrates.

Considering the many functions of FER (and other RKs) in different processes, it is highly likely that several phosphatases modify the same protein at different positions. Thus, binding FER seems to us not appropriate as a selection criterion for having a PP2C12-like activity.

Additional data shown that were not requested by the reviewers:

PP2C12 and PP2C15 have overlapping activity and PP2C15 negatively regulates BAK1 (Diao et al, referenced in the text). Hence, we felt that it would be good to rule out that BAK1 is involved in the aspects of root hair development that are described in this work. To this end, *lrx1 bak1* and *lrx1 rol23 bak1* double and triple mutants were produced, with both *bak1-4* and *bak1-5* alleles. The *bak1* mutations do not influence the *lrx1* or *lrx1 rol23* phenotypes. This analysis is shown in Fig. 3A,C and described on page 7, lanes 3-9).

We have included a new co-author to the manuscript (Álvaro D. Fernández-Fernández), who has done the live-cell imaging of FER-GFP and PP2C12-GFP shown in Appendix Fig. S6. All authors agree to include him as a co-author.

Finally, we have introduced a number of improvements (typos, labelling inconsistencies, better phrasing) that do not change the content of the manuscript. These changes can be seen in the document with the track-change mode, but are not listed here.

Dear Christoph,

We have now received re-review reports from two referees, which I have included below. As you will see, you have addressed their concerns satisfactorily. Before I can finally accept the manuscript, there are some remaining editorial points which need to be addressed. In this regard would you please:

- include five keywords,
 - as your study does not include external datasets, please state 'This study includes no data deposited in external repositories' in the Data Availability section,
 - include a 'Disclosure and competing interests statement',
 - include callouts in the manuscript for Fig. 2D, 3C, and 8A-B,
 - complete the general info table in the author checklist,
 - convert the appendix file to PDF format; the title page should contain "Appendix for The phosphatase PP2C12 is a negative player in LRX-RALF-FER-mediated cell wall integrity sensing " and a table of contents with the page numbers for the listed items,
 - provide source data for for Figures 6B and 9A, and for EV and/or appendix figures, ZIPping together all source data files,
 - please provide the summary figure at a higher resolution,
 - present the blots in Figure 7C with splice sites clearly marked with a line or with a white / black box around each blot.
 - provide exact p values in the legends of figures 1C, 2C, 3B-E; 4D, 6B, and 7D, F,
 - define box plots in terms of minima, maxima, centre, bounds of box and whiskers, and percentile in the legends of figures 3D, E,
 - define n in the legend of figure 4D,
 - define the nature of n in the legend of figure 4F,
 - define error bars in the legends of figures 4F 6B, 8B,
 - rename the "Materials and Methods" section as "Methods", and
- use the following section order: Title page - Abstract - Keywords - Introduction - Results - Discussion - Methods - Data Availability - Acknowledgements - Disclosure and Competing Interests Statement - References - Figure Legends - Table(s) - Expanded View Figure Legends.

I am looking forward to receiving your revised manuscript.

EMBO Press is an editorially independent publishing platform for the development of EMBO scientific publications.

Best wishes,

William

William Teale, PhD
Editor
The EMBO Journal
w.teale@embojournal.org

See also figure legend guidelines: <https://www.embojournal.org/page/journal/14602075/authorguide#figureformat>

- a point-by-point response to the referees' comments, with a detailed description of the changes made (as a word file).
- a word file of the manuscript text.
- individual production quality figure files (one file per figure)
- a complete author checklist, which you can download from our author guidelines (<https://www.embojournal.org/page/journal/14602075/authorguide>).
- Expanded View files (replacing Supplementary Information)

<https://www.embopress.org/page/journal/14602075/authorguide#expandedview>
- a Reagents and Tools Table as part of the Methods section, which can be downloaded from our author guidelines
(<https://www.embopress.org/page/journal/14602075/authorguide#structuredmethods>)

We realize that it is difficult to revise to a specific deadline. In the interest of protecting the conceptual advance provided by the work, we recommend a revision within 3 months (7th Dec 2025). Please discuss the revision progress ahead of this time with the editor if you require more time to complete the revisions. Use the link below to submit your revision:

Referee #1:

I thank the authors for carefully addressing the points raised. I am satisfied with their responses and recommend the manuscript for publication.

Referee #2:

I would like to thank the authors for addressing my comments. I have no further concerns or suggestions.

Dear William & editorial team,

My apologies for these things I didn't properly adjust. Below, you find a point-by-point response for the to-do list you sent. The manuscript file "EMBOJ-2024-119600R_Hou et al_CR4.docx" includes the changes to the manuscript that are described below (our answer is in bold).

- We couldn't find the keywords, could you please provide them below Abstract in your manuscript file?

My apologies, I thought I included them last time. This time, they are listed after the abstract.

- The following callouts in the manuscript text need correction as they are missing the word "Appendix": Fig. S7, Fig. S1, S4, S6. Please correct them

done

- You can remove the source data folder that combines all source data. The separately uploaded zip folders should stay as the source data files will be published alongside the figures

done

- If a reference has more than 10 authors, please use et al after the 10th author name in that reference

done

- Remove the Appendix Figure and Table legends from the manuscript file

All legends of the Appendix Figures have been removed, the Appendix Tables didn't have a legend. I have left the titles of the Appendix Figures and Appendix Tables, assuming that the title of the Figures and Tables are not part of the legend. I hope this is correct.

- Synopsis image: the uploaded image looks like cover art. What we need is a schematic summary figure that provides a sketch of the major findings (not a data image) in jpg, png or tiff format (with the exact width of 550 pixels and a height of not more than 600 pixels) that can be used as a visual synopsis on our website.

My apologies for being a bit slow in understanding the meaning of the synopsis image. I think the one I uploaded should be ok.

- We received bounced email alerts for the co-author Mona Häfliger - mona.haefliger@uzh.ch. Please either remove the author from the author list in the system and add her using a new email address or send us the new email address and we will update the author's account accordingly.

I have updated the author section with the active e-mail address of Mona Häfliger (haefligermona@gmail.com).

Dear Christoph,

I am pleased to inform you that your manuscript has been accepted for publication in the EMBO Journal.

Congratulations to you and your team!

Best wishes,

William

William Teale, PhD
Editor
The EMBO Journal
w.teale@embojournal.org
